# eXplainable Artificial Intelligence (XAI) for the identification of biologically relevant gene expression patterns in longitudinal human studies, insights from obesity research

**Augusto Anguita-Ruiz**[1,2,3]*, **Alberto Segura-Delgado**[4], **Rafael Alcalá**[4], **Concepción M. Aguilera**[1,2,3], **Jesús Alcalá-Fdez**[4]

**1** Department of Biochemistry and Molecular Biology II, Institute of Nutrition and Food Technology "José Mataix", Center of Biomedical Research, University of Granada, Granada, Spain, **2** Instituto de Investigación Biosanitaria ibs.GRANADA, Granada, Spain, **3** CIBEROBN (Physiopathology of Obesity and Nutrition), Instituto de Salud Carlos III (ISCIII), Madrid, Spain, **4** Department of Computer Science and Artificial Intelligence, University of Granada, Granada, Spain

* augustoanguitaruiz@gmail.com

**Data Availability Statement:** All relevant data are within the manuscript and its Supporting Information files.

## Abstract

Until date, several machine learning approaches have been proposed for the dynamic modeling of temporal omics data. Although they have yielded impressive results in terms of model accuracy and predictive ability, most of these applications are based on "Black-box" algorithms and more interpretable models have been claimed by the research community. The recent eXplainable Artificial Intelligence (XAI) revolution offers a solution for this issue, were rule-based approaches are highly suitable for explanatory purposes. The further integration of the data mining process along with functional-annotation and pathway analyses is an additional way towards more explanatory and biologically soundness models. In this paper, we present a novel rule-based XAI strategy (including pre-processing, knowledge-extraction and functional validation) for finding biologically relevant sequential patterns from longitudinal human gene expression data (GED). To illustrate the performance of our pipeline, we work on *in vivo* temporal GED collected within the course of a long-term dietary intervention in 57 subjects with obesity (GSE77962). As validation populations, we employ three independent datasets following the same experimental design. As a result, we validate primarily extracted gene patterns and prove the goodness of our strategy for the mining of biologically relevant gene-gene temporal relations. Our whole pipeline has been gathered under open-source software and could be easily extended to other human temporal GED applications.

## Author summary

Biological processes in humans are not single-gene based mechanisms, but complex systems controlled by regulatory interactions between thousands of genes. Within these gene regulatory networks, time-delay is a common phenomenon and genes interact each other

**Funding:** This work was supported by the Mapfre Foundation ("Research grants by Ignacio H. de Larramendi 2017") and by the Regional Government of Andalusia ("Plan Andaluz de investigación, desarrollo e innovación (2018), P18-RT-2248"). The authors also acknowledge the Institute of Health Carlos III for personal funding: Contratos i-PFIS: doctorados IIS-empresa en ciencias y tecnologías de la salud de la convocatoria 2017 de la Acción Estratégica en Salud 2013–2016, Project number: IFI17/00048. The funders had no role in study design, data collection and analysis, decision to publish, or preparation of the manuscript.

**Competing interests:** The authors have declared that no competing interests exist.

within a four-dimension space. Hence, to fully understand or to control biological processes we need to unravel the principles of gene-gene temporal interactions. Until date, several approaches based on Artificial Intelligence methods have tried to address this issue. Nevertheless, the research community has claimed for more interpretable and biologically meaningful models. Particularly, scientists claim for methods able to infer gene-gene temporal interactions that could be later validated with real-life experiments at the lab. The recent revolution known as "eXplainable Artificial Intelligence" offers a solution for this issue, where a range of highly interpretable and explicable models has become available. Many of these methods could be applied to temporal gene expression data in order to decipher mentioned temporal gene-gene relationships in humans. Here, we propose and validate a new pipeline analysis including an eXplainable artificial intelligence method for the identification of comprehensible gene-gene temporal relationships from human intervention studies. Our method has been validated in six datasets from obesity research (consisting of low calorie diets interventions), where it was able to extract meaningful gene-gene temporal interactions with relevance the etiology of the disease. The application of our pipeline to other type of human temporal gene profiles would greatly expand our knowledge for complex biological processes, with a special interest for drug clinical trials, in which identified gene-gene regulatory interactions could reveal new therapeutic targets.

## Introduction

Biological processes in humans are not single-gene based mechanisms, but complex systems controlled by regulatory interactions between thousands of genes. Within these gene regulatory networks, time-delay is a common phenomenon and genes interact each other within a four-dimension space [1]. That is to say, it may take a time since the product of a gene is generated, until it finally causes an effect on its target molecule. Some of the main sources of time-delay in gene regulation may include; 1) the action of gene expression co-activators or co-repressors, 2) the influence of external environmental factors, and 3) the natural self-degradation of messenger RNA and proteins in cells. Time-delayed gene regulation is especially present in long-term interventions, in which changes in gene expression reflect the response of genes to external factors and may cause subsequent changes on the expression of other genes.

DNA microarray technology has provided a powerful vehicle for exploring biological processes on the genomic scale. In spite of it, in most of the genome scans performed to date, the effects of each gene on the trait of interest have been interrogated one at a time; thus presenting a limited throughput to get the overall picture of gene networks and their temporal relations. Unsupervised methods implemented in conventional microarray software (such as clustering solutions) have also failed in the discovery of network phenomena, since genes can participate in more than one network all at once. As a result, there is not a clear picture of the dynamic trends in gene-gene interactions and much of the heritability of complex human traits remains unexplained, a phenomenon termed as the "missing heritability" problem [2].

The creation of public functional genomics data repositories has enabled a huge amount of genome-wide expression profiles become available to the scientific community [3]. Among available datasets, the recent increase of massive temporal microarray experiments (such as clinical and dietary long-term interventions) open up new opportunities to uncover time-delayed gene-gene relationships. Several machine learning (ML) approaches have been proven very effective for extracting associations between different genes, highlighting Boolean models,

Bayesian networks and Neural networks [4–7]. Due to their great predictive ability, these ML methods have been widely used in this and other field applications. Nevertheless, despite yielding impressive results, most of these techniques output unintelligible and complex gene networks, and can not explain how they arrive at specific decisions (which is known as the "black box" problem) [8–10]. For scientists to trust they must first understand what machines do, since in many cases it is not so much what an algorithm predicts but the relationships it establishes and how it predicts it. This is especially important in gene networking, where one of the main concerns of biologists is how to translate inferred networks into particular hypotheses that can be tested with real-life experiments. On this sense, there is a recent increased need to provide ML models with more interpretability and explicability, giving rise to what it is known as eXplainable Artificial Intelligence (XAI) [8,10,11]. As one of the most naturally interpretable and popular knowledge discovery techniques, association rule mining has become a highly relevant technique within the XAI revolution, being able to generate practical knowledge understandable from the point of view of human experts [12,13]. Practical knowledge in association rule mining is extracted in the form of association rules and it refers to concrete relationships between the elements of a database. Association rules constitute representations with the form of X → Y, which means that when X occurs it is likely that Y also occurs. Due to its natural explicability, association rule mining methods have emerged as an excellent choice for the data mining of complex biological datasets in humans [14]. On this sense, they have been successfully applied to gene expression data (GED) in order to represent how the expression of one (several) gene(s) may be linked or associated with the expression of a different set of genes (gene-gene interactions discovery) [14,15].

Although interesting insights have been derived from the application of association rule mining to GED, previously mentioned time dependencies between associated genes cannot be modelled making use of conventional association rule mining methods. To face this problem, sequential rule mining (SRM) algorithms could be used instead. SRM algorithms are intended to discover interesting sequential relationships between the elements of a sequence database, in which the data are represented sequentially (e.g. time ordered or spatially localized). The concept of a sequential rule is similar to that of association rule but, in this case, X must appear before Y according to the sequential ordering criterion of the database [13,16]. By way of example, sequential rules that can be extracted from the application of SRM to temporal microarray designs has the next form; [gene A↑, gene B↓] → (time delay) [gene C↑, gene D↑, gene E↑], which represents that the upregulation of gene A and the significant repression of gene B are followed by (or cause) a significant upregulation of genes C, D and E after a given time delay.

Until date, there has been only one adaptation of SRM methods to temporal microarray data [15], consisting of an *Ad-hoc* application for *in vitro* time-series GED in *Saccharomyces cerevisiae*. Referred to as temporal ARM (TARM), the employed method is based on the conventional association rule mining algorithm "*Apriori*" and has been exclusively designed to work with GED derived from yeast (composed of 799 genes evaluated during five transcriptional time points in the same culture). This method builds a sequence database conformed by a single sequence of events (i.e. same culture) by converting each continuous gene expression value into a discrete item by time interval (upregulation, downregulation, or none). Then, TARM is able to identify concrete and understandable temporal causal relations among genes with relevance in the yeast cell cycle.

Besides previous approach, it is also remarkable a more recent work published by Liu et al. (2013) [17], in which a sequential pattern mining algorithm is proposed for the identification of temporal co-expression networks from *in vitro* human data. Sequential pattern mining algorithms belong to the ML branch of frequent pattern mining and could be considered as a

simpler version of SRM (sequential rule mining). The main drawback of sequential pattern mining algorithms, in comparison to SRM methods, is that they find sequential patterns that appear frequently in a sequence database but without generating sequential rules from them. Thus, they are unable of establishing causal relationships between items, and their resulting sequential patterns could be misleading in certain occasions, especially in the presence of very frequent elements in a sequence database [16]. For these reasons, although the Liu et al. (2013) [17] approach interestingly moves forward from yeast models into *in vitro* human data, and is based on a highly interpretable ML method, it still presents some drawbacks for its fully application in the modeling of temporal gene networks.

As Liu et al. (2013) [17], many other researchers have also explored co-expression for the reconstruction of gene networks in humans (although not from a pattern mining approach). These methods have become thereby the gold-standard when studying a microarray experiment from a systems biology perspective [4,5,18,19]. In the case of time course genomics experiments, most of conducted co-expression approaches have been based in clustering analyses [20]. These temporal co-expression approaches are generally based on similarity or correlation or distance measures for the identification of groups of genes with 'similar' temporal patterns of expression, and reveal hidden patterns in the original data by transforming raw temporal data into logically structured, clustered, and interconnected graphs [18,19]. Co-expression graphs can be visualized with nodes representing genes, and with edges indicating interactions, and have helped to understand how genes interact each other within the context of an integrated and global biological network. Nevertheless, despite the widespread use of these approaches [4], there are some drawbacks and limitations remaining for their application in the inference of causal gene-gene relationships:

- Co-expression networks are good to study the general interactome of an organism (free-scale network topology), but their results make hard to infer particular details such as the causal direction or the importance of each individual interaction within the whole network [20]. Thus, they may hinder the translation of inferred networks into particular hypotheses that can be tested in wet-lab experiments. On the contrary, SRM results (in the form of individual rules for each interaction) allow a concrete quality evaluation for each relationship and an easy biological interpretability of findings, which is crucial to demonstrate that a gene network is functionally meaningful, and not just biostatistical fluke [4].

- Co-expression is a very strict assumption for the extraction of gene-gene interactions from time course data [21]. That is to say, co-expression networks with temporal GED generally do not include the time order information in graphs, and they are not capable of detecting positive, negative and time-lagged gene correlations at the same time [20]. However, in living systems, gene regulations can be positive or negative possibly with time lags, and may also not span all conditions or time points. For example, some of the target genes could have a negative feedback loop and could block their own expression, which could explain fast transient dynamic changes, while other target genes could have a positive feedback loop and therefore maintain gene expression longer. Additional regulation could happen after a longer time or very fast without protein translation, i.e. by the action of functional large noncoding RNAs. All these kind of phenomena, which are missed by most of co-expression approaches, could be captured by an appropriate SRM approach.

Given all these concerns and the interesting properties of SRM methods (e.g. existence of statistical quality measures by rule, possibility of functional validation by interaction, inclusion of causality or sequential order information, discovery of complex temporal regulatory

phenomena . . .), SRM approaches are presented as an alternative of great interest and interpretability against temporal co-expression clustering methods when inferring gene-gene temporal relations.

Unfortunately, as far as we concern, the work of Nam et al. (2009) [15] in *Sacharomyces cerevisiae* is the only SRM approach developed to the moment and it constitutes no more than an *Ad-hoc* application for *in vitro* experiments (whose extension to *in vivo* GED would elicits challenges that could not be solved with simple algorithm modifications). The application of these methods to human temporal gene profiles otherwise would greatly expand our knowledge for complex biological processes, with a special interest for long-term interventions (such as clinical trials), in which identified gene-gene regulatory interactions could reveal promising and new therapeutic targets [22,23]. Among the main issues that may have prevented the adaptation of SRM for temporal gene networking in *in vivo* human data we can highlight:

1. **The high dimensionality of human gene expression microarrays**. With more than 30.000 probes under study in conventional human microarray platforms, the volume of the search space is so big that any available data will become sparse (especially in the case of clinical trials where sample sizes are barely composed of a few tens). The low number of temporal records that are normally assessed in this kind of interventions (rarely more than four) further worsens this sparsity. Within this context, most of ML approaches will thus present a detrimental performance and reliable results can be obtained only if the study sample size is exponentially increased or if effective 'feature-selection' methods are employed prior to analysis for dimensionality reduction.

2. **The lack of gene expression discretization methods for *in vivo* datasets**. Most of available SRM algorithms require categorical data as input to perform inference. Thus, the selection of an appropriate discretization strategy is a key step for a successful performance. A wide range of *in vitro* discretization methods has been recently revised and gathered under open-source software [24]. Of note, performance of these methods have shown strong dependence on the particularities of each biological problem. Regarding human GED, there are few issues to take into account before performing discretization, including not only the fact of having multiple sequences but also the great variability between (and within) subjects, tissues and conditions. Considering all this, the extension of existing *in vitro* discretization strategies to humans is not a trivial issue and new approaches should be proposed.

3. **The problem of mining sequential rules common to multiple sequences**. Contrary to what happen *in vitro*, in human or animal experiments the "subject variability" is the main issue to address, so that databases include multiple sequences instead of a single one. That is to say, we pass from single-sequence experiments where only one microarray is conducted in the culture by time record, to an experimental framework with N > 1 gene expression profiles evaluated at each data point (being N the number of subjects under study). Since classical SRM methods have been originally intended for mining sequential rules in a single sequence of events, gene networking in human temporal microarrays will require the adaptation of more modern and specific SRM algorithms.

4. **The need for functional validation of results**. As it is well known, when facing high-dimensionality data with low sample sizes, data mining methods may yield results which seem to be significant; but which do not actually represent real behaviors of the dataset. Regarding the case of SRM and its application to GED, this problem is presented in the form of too many output rules even after pruning quality application, which can reach the order of thousands. In such cases, the extracted SRM-based gene networks will represent a

chaotic set of "potential" interactions whose biological interpretation will become a serious challenge for biologists.

In this paper, we present a three-stage and rule-based XAI strategy (including pre-processing, knowledge-extraction and functional validation) for finding biologically relevant sequential rules from longitudinal human GED. Particularly, our strategy involves the proposal of an improved version of the well-known SRM algorithm CMRules [25] in order to mine time-delayed gene relationships from *in vivo* human temporal microarray data. Furthermore, we not only adapt the CMRules algorithm to the specific GED problem but also propose a full-detailed data pre- and post-processing pipeline that solve previously mentioned human data limitations and increase model explicability. As a result, our methodology is able to generate temporal gene expression networks in long-term human interventions. The proposed pre- and post-processing pipeline could be briefly summarized in the following key aspects:

- First, the initial number of probes is reduced to those differentially expressed by time interval and experimental condition. This way, we simplify the experimental problem and reduce the search space, further favoring a better performance of the algorithm.

- Secondly, we propose a new discretization approach for the conversion of continuous gene expression values into discrete categories representing temporal changes in gene expression. Based on signal log ratios by gene and time interval, this discretization strategy maps data from a vast spectrum of numeric gene expression values into three discrete categories. Therefore, it can be viewed as a secondary data dimensionality reduction technique in favor of model explicability.

- Third, we apply the SRM algorithm CMRules to the discretized dataset and generate sequential rules from it with the form of [gene A↑, gene B↓] → (time delay) [gene C↑, gene D↑, gene E↑]. Each rule is assessed in terms of quality and robustness by means of five interestingness or quality metrics as described in the *method section*.

- After the knowledge-extraction stage, we propose the integration of output gene rules along with external biological resources such as functional annotation and gene regulation databases. Three well-known and reliable biological resources (GO, KEGG and TRRUST databases) are consulted in order to compute five new biological quality measures by-rule. Through this strategy, each interaction result is biologically pruned and placed within the context of those molecular systems that commonly underlie gene-gene interactions in humans (e.g. Transcription factor (TF)-target gene regulatory relationships [26]).

- Finally, we propose data visualization for the joint representation of gene patterns and all accessed biological information. By means of hierarchical edge bundling visualization methods, we concentrate a lot of information in a single shot and facilitate the identification of a finite set of genes composing a good quality network.

The whole pipeline of our proposal is illustrated in the S1 Fig and has been fully detailed in the *method section*. The strategy presented in this work (with special relevance of the adopted ML algorithm, and the functional validation and graphical representation of results) constitute a value proposal whose main objective is to increase model *eXplainability* and to help biologists understanding extracted gene interactions. In this way, we pretend to move away from the black box concept that is usually adopted in most of the current artificial intelligence (AI) omics applications [27], and to provide researchers with a great power to discern between random and causal gene relationships. Our whole pipeline and the SRM adaptation have been gathered as open-source software in the public hosting GitHub (https://github.com/

AugustoAnguita/GeneSeqRules) and could be easily extended to other temporal GED applications. At the end, we hope this proposal becomes a helpful strategy for the identification of comprehensible genetic interactions in long-term human interventions, with special interest for the discovery of novel therapeutic targets in clinical datasets.

Since our method is the first application of a SRM strategy for the extraction of gene interactions in longitudinal human *in vivo* experiments, there is not currently a benchmark tool that we could use to compare the performance of our pipeline with. At least, not without implementing algorithm modifications in such comparison methods. In spite of it, from the biomedical perspective the real challenge issue when inferring gene networks is their reliability for avoiding false discovery as well as their reproducibility across different patient cohorts. For this reason, we decided to validate our approach in two alternative ways: 1) First, we applied our methodology to an example dataset and give the derived results to a group of field-experts in order them to evaluate the usability of inferred networks for the generation of particular gene-gene interaction hypotheses; and 2) We repeated the application of the full pipeline to three additional datasets, following the same experimental design than the discovery sample, and mined results looking for replication patterns across studies. As a result, we validated some of the primarily extracted gene patterns and thus proved the goodness of our strategy for the mining of biologically relevant gene-gene temporal relations (see results *section*). Full details for the evaluation guidelines committed by field-experts during the interpretation of model results have been addressed in the *method section*.

Main topics covered in this paper include: 1) Preliminary concepts in ARM. 2) Methodological description of the proposed pipeline (including pre-processing, Knowledge-extraction stage and Functional validation of results), 3) Description of the research problem and employed datasets, 4) Results description, where we evaluate the performance of our pipeline in terms of the insights extracted from a discovery sample and their validation in independent cohorts and 5) Discussion section, where we deepen the goodness of our proposal and list some drawbacks and challenges to be faced in future applications.

## Methods

### Preliminary: Association rules and sequential rules

The concept of association rules was first proposed by Agrawal et al (1993) [12] as a market basket analysis tool in order to discover what items are bought together during a supermarket purchase. Many algorithms for mining association rules and other that extend the concept of association rule mining have been proposed so far to extract useful knowledge from different types of transactional datasets (*T*). As previously mentioned, association rules have the form of *Left Hand Side* (LHS) → *Right Hand Side* (RHS), where LHS and RHS are sets of items, and it represents that the RHS set being likely to occur whenever the LHS set occurs. Interestingly, association rules move forward from the simpler concept of frequent patterns and allows the opportunity to uncover true causal relationships between items [13]. In the field of gene networking, an example of transactional dataset would be a subset of individuals belonging to an experimental condition; where each individual from the subset would be considered as a transaction of the database, and each gene expression event for that particular individual (e.g. gene A↑, gene B↓, gene C↑, gene D↑, gene E↑...) would be considered as an item composing that particular transaction. Support and confidence are the most common measures used to assess association rules' quality, both of them based on the support of an itemset. In the previously introduced example, an itemset would refer to any combination of items from the database (e.g. gene B↓ & gene C↑), being also possible the fact of an itemset composed by only one item.

In association rule mining, the support for an itemset *I* is defined as:

$$\text{SUP}(I) = \frac{|\{e \in \mathrm{T} | \mathrm{I} \in e\}|}{|T|} \tag{1}$$

where the numerator is the number of examples (*t*) in the dataset *T* covered by the itemset *I*, and | *T* | is the total number of examples in the dataset. Thus, the support and confidence for a rule LHS → RHS are defined as

$$support(LHS \rightarrow RHS) = SUP(LHS \; U \; RHS) \tag{2}$$

$$confidence(LHS \rightarrow RHS) = \frac{SUP(LHS \; U \; RHS)}{SUP(LHS)} \tag{3}$$

In other words, support could be viewed as the percentage of transactions where the rule holds, and confidence as the conditional probability of RHS with respect to LHS (i.e. the relative cardinality of RHS with respect to LHS). The classic techniques for mining association rules attempt to discover rules whose support and confidence are greater than certain user-defined thresholds called minimum support (minSup) and minimum confidence (minConf). However, several authors have pointed out some drawbacks of this framework that lead to find many misleading rules [28]: 1) First, the confidence measure is not able to identify statistical independence or negative dependence between LHS and RHS, mainly due to the fact that the RHS support is not taken into account during the computing process, and 2) Second, itemsets with very high support will be a source of misleading rules because they exist in most of the examples (transactions) and therefore any itemset could seem to be a good predictor of the presence of the high-support itemset. The following example is from [29] and it illustrated very well previous misleading behaviors: in the CENSUS database of 1990, the rule "past active duty in military ⇒ no service in Vietnam" has a very high confidence of 0.9. This rule suggests that knowing that a person served in military we should believe that he/she did not serve in Vietnam. However, the itemset "no service in Vietnam" has a support over 95%, so in fact the probability that a person did not serve in Vietnam decreases (from 95% to 90%) when we know he/she served in military, and hence the association is negative. Clearly, this rule is misleading.

To face these problems, researchers have proposed additional quality metrics by rule and have introduced the concept of "very strong association rules" [30], which are of great aid for the selection and ranking of rules according to their potential causality and coherence. Next, we briefly describe some of the additional metrics that have been used in this paper as well as introduce the framework of "very strong association rules".

The conviction [29] measure analyzes the dependence between LHS and ¬RHS, where ¬RHS means the absence of RHS. Its domain is [0,∞), where values less than one represent negative dependence, a value of one represents independence, and values higher than one represent positive dependence. Conviction for a rule LHS → RHS is defined as

$$conviction(LHS \rightarrow RHS) = \frac{SUP(LHS)SUP(\neg RHS)}{SUP(LHS \neg RHS)} \tag{4}$$

The lift [31] measure represents the ratio between the confidence of the rule and the expected confidence of the rule. As with conviction, its domain is [0,∞), where values less than one imply negative dependence, one implies independence, and values higher than one

imply positive dependence. Lift for a rule LHS → RHS is defined as

$$lift(LHS \rightarrow RHS) = \frac{SUP(LHS\ U\ RHS)}{SUP(LHS) * SUP(RHS)} \tag{5}$$

The certainty factor (CF) [32] is interpreted as a measure of variation of the probability that RHS is in a transaction when we consider only those transactions where the LHS is present. Its domain is [–1,1], where values less than zero represent negative dependence, zero represents independence, and values higher than zero represent positive dependence. CF for a rule LHS → RHS is defined in three ways depending on whether the confidence is less than, greater or equal to SUP(RHS):

if confidence(LHS → RHS) > SUP(RHS)

$$\frac{confidence(LHS \rightarrow RHS) - SUP(RHS)}{1 - SUP(RHS)} \tag{6}$$

if confidence(LHS → RHS) < SUP(RHS)

$$\frac{confidence(LHS \rightarrow RHS) - SUP(RHS)}{SUP(RHS)} \tag{7}$$

Otherwise is 0.

Some of presented metrics, such as the CF, have been further employed to create a framework intended to make easier the discovery of those patterns known as "very strong association rules" [30]. Particularly, a rule will be considered as very strong (and thus not a misleading relationship) if it fulfills the following conditions (Support > minSup, Not(Support) > (1- minSup) and CF > 0). The concept of very strong rule is very intuitive, since it is based on the logical equivalence between a rule and its counter-reciprocal, and it captures the idea that, since both rules are equivalent, finding evidence of both in data enforces our belief that the rule is important.

Although association rule mining methods and all presented metrics have shown a good ability to mine hidden relationships in many different domains (such as genetics [14], biomedicine [33], and so on), these methods are aimed at analyzing data where the sequential ordering of events is not taken into account. Consequently, when such techniques are applied on data following a specific time or sequential ordering criterion, this information will be ignored. This situation may result in the failure of association rule mining methods to extract interesting knowledge from the data, or in the extraction of knowledge that may not be useful for the experts. Otherwise, in many domains, the ordering of events or elements is important and, particularly in genetics, the temporal information is especially critical for the understanding of the regulatory mechanisms of biological processes. SRM algorithms [16] have proven to be an interesting method for discovering sequential relationships between the elements of a sequence database (in which the data are represented sequentially or time ordered). Whenever the time dimension appears, SRM approaches will present a greater predictive and descriptive power than conventional association rule mining algorithms and will provide an additional degree of interestingness. Furthermore, SRM resolve an important limitation of the previously introduced simpler technique sequential pattern mining, since a sequence pattern may appear frequently in a sequence database but may have a very low confidence (which makes it therefore not useful for the identification of causal relationships). The concept of a sequential rule that can be extracted from SRM is similar to that of association rule except that it is required that LHS must appear before RHS. Previously mentioned quality measures (support and confidence) are also employed in SRM in order to evaluate the interestingness of each mined rule.

In SRM, rules are extracted from a sequence database. Recovering the same previous example of gene networking, an example of sequence database could be a subset of individuals belonging to a long-term intervention (with more than two time point records available) in which each individual from the subset would be considered as a sequence of the database, and each gene expression change event for a particular individual and a particular time interval (e.g. gene A↑ from T1 to T2, gene B↓ from T1 to T2, gene C↑ from T2 to T3...) would be considered as items composing that particular sequence. In SRM, the introduced basic quality metrics by rule are defined as sequential support (seqSup) and sequential confidence (seqConf)

$$\text{seqSup}(\text{LHS} \rightarrow \text{RHS}) = \frac{sup(LHS \rightarrow RHS)}{|SD|} \tag{8}$$

$$\text{seqConf}(\text{LHS} \rightarrow \text{RHS}) = \frac{sup(LHS \rightarrow RHS)}{sup(LHS)} \tag{9}$$

Here, the |SD| refers to the total number of sequences in the sequence database. The element *sup(LHS→RHS)* refers to the number of sequences from the sequence database in which all the items of LHS appear before all the items of RHS (note that items within LHS (or RHS) do not need to be in the same sequence nor temporal order within each sequence). The notation *sup(LHS)* refers to the number of sequences that contains LHS. In addition to seqConf and seqSup, the rest of previously introduced association rule mining quality metrics (such as conviction, lift and CF) have also their extension in SRM based in the definition of sequential support, keeping their original meaning and domains. All of them have been incorporated by our methodology for the mining of temporal sequential patterns in GED and allow practitioners a quick identification of the robustness of each extracted pattern from a frequentist perspective.

## Pre-processing stage: Feature selection and data discretization

As input files, our methodology receives raw fluorescence intensity signals (one *.cel* file per subject and time point), and perform a transformation into the form of an *N x M* matrix of gene expression values, where the *N* rows correspond to subjects under study and the *M* columns correspond to evaluated gene probes. All available time records are then merged into a single primary database so that each subject under study will present as many consecutive entries in the database (Long-format) as temporal points exist in the experiment (corresponding to the subject's gene expression profile at each time point). A primary quality control process is then conducted following straightforward pre-processing analyses in transcriptomics (generating chip pseudo-images, histograms of $log_2$(intensities) and MA-plots). Finally, all microarray fluorescence signals are normalized together by means of the robust multichip average (RMA) method and probes are annotated according to the latest released version of the "*org.Hs.eg.db*" database [34].

When dealing with Affymetrix microarray technologies, the huge number of probes available in platforms (often around 33.000) may induce an exponential growth of the search space, so that the knowledge-extraction process (independently of the ML method used) will become a difficult and complex task exceeding the processing capability of conventional systems. In order to solve this problem, prior to knowledge-extraction, our methodology includes a feature-selection step in which the number of probes is reduced according to the differentially expressed (DE) genes by time interval and experimental condition. Given a longitudinal GED experiment, our method identifies DE probes by assessing the changes in gene expression during each period of intervention. A probe will be selected for downstream analyses when its Bonferroni-adjusted P-value is < 0.05 and the associated *$Log_2$(FoldChange)* (which is also known as the signal log ratio) is > = 1 or < = -1 in a paired t-test with Bayesian correction.

After feature-selection, as the second main-step of the data pre-processing process, gene expression data discretization is also incorporated in our pipeline. Data discretization is a technique commonly employed in computer science that has been successfully applied to GED applications [24]. Here, the main motivation behind the application of GED discretization is allowing the use of ML algorithms, such as SRM, that requires discrete data as an input for the inference of biological knowledge. Nevertheless, there are many other advantages that arise from data discretization in genetics; 1) Discrete states favor the inference of qualitative models [35], which are of special importance in terms of model explicability. The explicability improvement in qualitative models is achieved due to the fact that for scientists, discrete values are easier to understand, use and explain than continuous values [35,36]. On this matter, GED discretization can be viewed as a secondary data pre-processing technique that move ML approaches closer to the XAI trend. 2) Another advantage emerging from GED discretization is the homogenization of different datasets in terms of interpretability. If the same semantics is used for the discretization of heterogeneous datasets, their results will be more easily comparable and the application of the same ML method to all of them will be a more straightforward and affordable task [37]. 3) Finally, the learning process from discrete data is more efficient and effective (requiring a reduced amount of data and yielding more compact and shorter results) [36]. Therefore, this step not only allows the inference of large-size models with a higher speed of analysis but also facilitate a significant portion of the biological and technical noise presented in the raw data to be absorbed, which may indirectly lead to a better model accuracy.

A wide range of *in vitro* GED discretization methods have been recently revised [24]. In our method, we adopted an unsupervised discretization approach based on expression variations between time points and adapted it to the problem of *in vivo* human data. For that purpose, continuous gene expression values from the filtered gene expression primary matrix ($N \, x \, M$) are transformed into three discrete categories (items) representing changes in gene expression. This approach gives a discretized matrix A of M probes and N—1 conditions, in which each probe by time interval may have one of three discrete states: 2, 1 and 0, meaning 'increase', 'decrease' and 'nochange' respectively. For the assignation of these discrete states by probe and time interval, each experimental condition of a dataset is considered separately as the 'data scope' framework and the next criteria are considered:

- For probes ik showing a positive signal log ratio in the previous DE analyses (feature-selection step):

  If $log_2(FoldChange)_{ikj} > log_2(FoldChange)_{ikJ}$ Then assign the label 'Upregulation'. Otherwise type 'No change'

- For probes ik showing a negative signal log ratio in previous DE analyses (feature-selection step):

  If $log_2(FoldChange)_{ikj} < log_2(FoldChange)_{ikJ}$ Then assign the label 'Downregulation'. Otherwise type 'No change'

  Where the term $log_2(FoldChange)_{ikj}$ refers to the signal log ratio (i.e. change in the gene expression) for the *i* probe, in the *k* time interval and the *j* subject from the group or dataset under study, and the term $log_2(FoldChange)_{ikJ}$ otherwise refers to the mean signal log ratio for that particular probe *i*, in the *k* time interval in all subjects from the group or dataset under study J.

The discretization approach adopted in this work can be viewed as an extension of two previously described *in vitro* temporal methods that have been successfully applied for the reconstruction of gene regulatory networks [24,35,38]. The main motivation behind its choice is the

fact that, when facing temporal GED, discretization methods based on transitions between time-points have been shown to obtain better results than those using absolute values [39]. Moreover, in our case, the use of a standardized measure of gene expression change (such is the Signal Log Ratio (SLR)) is a more sophisticated approach than the employment of simple differences between values. For example, the use of logs in the analysis eliminates difficulties caused by one very high data point in the set masking information from lower valued data points. On the other hand, base 2 is further used as the log scale, therefore a SLR of 1 represents a two-fold increase in abundance of an mRNA and a value of –1 represents a two-fold reduction in transcript expression. Finally, the use of the mean SLR ($log_2(FoldChange)_{ikJ}$) as the threshold for discretization allowed us to exclusively focus on general change trends in the dataset, which most likely will be consequence of the intervention conducted in the cohort and not particular gene expression changes due to individual's idiosyncrasy. Mapping data from a vast spectrum of numeric gene expression values to a reduced subset of three discrete states, this type of discretization could further be viewed as a secondary data reduction technique in favor of algorithm efficiency and *eXplainability*.

## Knowledge-extraction stage: Extension of the sequential rule mining method CMRules

Once the dataset is properly formatted, for performing knowledge-extraction, our method includes an adaptation of the well-known SRM algorithm CMRules [25]. Contrarily to other SRM methods that can only discover sequential rules in a single sequence of events, CMRules is able to mine sequential rules in several sequences, which makes it an excellent choice for dealing with human microarray temporal data. Furthermore, CMRules proposes a more relaxed definition of sequential rule with unordered events within each (LHS/RHS) part of the rule. Thanks to that, it presents a great ability to recognize the fact that similar rules can describe a same phenomenon; thus avoiding undesirable losses of information. Moreover, this characteristic also allows the method to detect some particularities of gene temporal interactions in human, such is the fact that gene regulations may not span all conditions or time points or that they could not occur at the same time-delay interval in all subjects from the intervention. Generally, CMRules starts applying the classic association rule mining method *Apriori* for extracting association rules without taking into account the temporal information. Next, the sequential support of each extracted rule is calculated in order to generate sequential rules from them. A detailed description of CMRules algorithm is presented in S2 Fig and has been reported elsewhere [25]. Besides the classical presented metrics (sequential confidence and support), we further computed more sophisticated sequential quality measures by rule as previously introduced (sequential lift, CF and conviction). Altogether included quality metrics allow practitioners a quick evaluation of the robustness of each extracted rule.

In order to deal with a common particularity of long-term interventions in humans (which is the fact of having two different intervention groups usually consisting on a placebo and a treatment group), a particular extension was implemented in the final step of the algorithm. This modification allows the user to choose that CMRules only show those sequential rules which besides fulfilling the condition (seqSup(r) > minimum sequential support (MinSeqSup) & seqConf(r) > minimum sequential confidence (MinSeqConf)), are further exclusive of each experimental group. These would be thereby all rules that assert the condition (seqSup(r) > MinSeqSup & seqConf(r) > MinSeqConf) in one experimental group but not in the other, and viceversa. Thanks to that, this extension of our method is presented as an excellent choice for the study of human clinical trials in which researchers are commonly interested in the discovery of gene-gene signatures activated by a specific treatment but not by a placebo.

## Functional validation stage: New biological quality measures by rule and visualization tool

Previous works have demonstrated that the integration of external biological resources within the gene networking process is a helpful strategy that improves model *eXplainability* and helps biologists to better understand genes and their complex relationships [14]. In recent years, a great variety of external databases containing biological knowledge has become available. Among the most robust and reliable ones, it highlights those containing information relative to gene function, protein localization and molecular interaction (e.g. the gene ontology (GO) project [40] and the Kyoto Encyclopedia of Genes and Genomes (KEGG) [41]). The GO project is an annotation database that provides a structured and controlled vocabulary to describe gene and gene product attributes in multiple organisms according to three different categories or ontologies (cellular component, molecular function and biological process). The KEGG database, on the other hand, is a bioinformatics resource that integrates current knowledge on molecular interaction networks, cellular pathways and functional information of genes and their products. Both GO and KEGG resources have been successfully employed in previous microarray ARM analyses aiding biological explanation to the extracted associated gene sets [14].

One of the main mechanisms controlling gene expression changes in living organisms is the action of gene-specific TFs. By binding to a particular DNA sequence, TFs regulate the—turn on and off—of target genes in order to make sure that they are expressed in the right cell and at the right time. Understanding the basis of genetic interactions between TFs and their targets is therefore likely to be important for the understanding of time-delayed gene regulatory relations in humans. For this reason, we propose the use of an additional biological database, known as TRRUST [26], which includes information relative to transcriptional regulatory relationships between hundreds of genes. The current version of the TRRUST database (version 2) contains 8.444 and 6.552 TF-target regulatory relationships for 800 human and 828 mouse TFs respectively. Especially for the application of SRM to temporal microarray data, the integration of TFs information is indispensable if we want to understand the complex gene relationships that are illustrated in the form of sequential rules.

In this paper, we propose the incorporation of these three well-known biological data resources (TRRUST, KEGG and GO) in order to evaluate extracted rules within a biological framework and to aid explicability to output models. For that purpose, we compute five new by-rule quality measures named "Biological Process (BP)", "Molecular Function (MF)", "Cellular Compartment (CC)", "Signaling Pathway (SP)" and "Transcription Factor (TF)"; that respectively integrate annotation terms from the three categories of the GO project, metabolic pathway annotations from the KEGG resource and transcriptional relationships from the TRRUST database. The computing process for each measure differs according to their biological meaning and the external resource in which their are based on. The first four measures (BP, MF, CC and SP) constitute rankings computed on the identical matches (between LHS and RHS items) that each rule presents for pathway identifiers and GO-terms annotated at the gene level in the previously mentioned GO and KEGG resources. Therefore, a lower resulting value in these biological metrics for a certain rule will indicate that the rule is a good candidate for representing a potentially causal and biologically relevant gene interaction. For each of these four quality measures, the final ranking-score by rule is computed based on the type and number of reported matches between LHS and RHS. According to the encountered types of matches, rules will be allocated into five different categories and will receive different number of points (Fig 1). Based on these definitions, the final ranking score for a given rule is

computed as follows:

$$MEASURE\ RS(r) = CAT(r) + \left(1 - \frac{NP(r)}{(\max(NP(i)), \forall i \in Cat(r)) + 1}\right) \qquad (10)$$

Where *MEASURE* refers either to "BP", "MF", "CC" or "SP", *RS(r)* refers to the ranking score obtained for the rule under study, *CAT(r)* refers to the designated rule category, *NP(r)* refers to the number of points assigned to the rule under study and max(*NP(i)*), $\forall_i \in Cat(r)$ corresponds to the maximum number of points that have been assigned to a rule from the same category. Specific details for the calculation of *NP(r)* and for the designation of a rule category *CAT(r)* are illustrated in Fig 1.

On the other hand, the biological quality measure TF constitutes a range of four possible values by rules (0, 1, 2 or 3) which are assigned according to the TF-target gene regulatory information hosted in the TRRUST database. The computing process for the TF measure is slightly different from the previous ones and is performed as follows; if at least one LHS gene from the evaluated rule has been reported as a validated TF in the TRRUST resource, then assign *1* point to the rule. Otherwise, assign *0* points. If the first condition has been fulfilled, then check if any of the RHS items (genes) from the rule has been presented in the TRRUST database as one of the previously-identified TF confirmed targets. In such case, assign *2* points to the rule. Otherwise, assign *0* points. If the previous condition has been further fulfilled, then check for the type of relationship that is reported in the TRRUST database for both TF and target genes (upregulation, downregulation or unknown). In the case of match between the relationship illustrated by the sequential rule (upregulation or downregulation of the target) and the information hosted in the TRRUST database, then *3* points are assigned to the rule.

The choice of that particular procedure, against other available standard GO-similarity measurements [42], was argued in the fact that we needed an evaluation method, based on categories, with the ability to discern the quality of a rule regardless of the items conforming it. If this were not the case, the rules with the highest number of antecedent/consequent elements would always have a higher score by the simple chance of coinciding in GO terms due to the greater number of genes composing them. On the contrary, with our heuristic approach, the score is adjusted by the number of items (from the total elements that constitute the rule) that share a specific GO term. Thanks to that, the rules in which all its elements share the same GO term will be identified as more robust than others, which although perhaps share a greater number of GO terms among all its antecedent elements, do not share any GO between all LHS and RHS items.

The computing process of these new five biological quality measures by rule has been implemented in python environment and can be directly applied to any output generated from the application of our ML method to temporal GED. The software requires a *.pmml* file as input (containing extracted rules with CMRules) and will output a same format file but with the new biological quality measures computed by rule. The software will also output a secondary file with *.xml* extension in which all the matches between the items of a rule are fully detailed (including information for genes and GO or KEGG terms composing each match of the rule). This software constitutes therefore a useful tool for the evaluation of extracted rules within the context of different human molecular systems and effectively complements the action of previously introduced classical SRM quality metrics. The main utility of the introduced biological quality measures has been illustrated in Fig 2, were we have tried to show their capacity to distinguish between true potential causal gene-gene interactions and those representing spurious biostatistical fluke. This figure represents how, although the statistical quality metrics by rule have been shown effective for detecting robust associations between

# RANKED CATEGORIES by measure

## 01
### Range of values [1,2)

## Top Category

A rule will belong to this category if **all** its *LHS* items share <u>the same term</u> with one (or more) *RHS* items.

| A | B | | C | D |
|---|---|---|---|---|
| GO:01 | GO:01 | → | GO:03 | GO:09 |
| | GO:05 | | GO:01 | |

Each time this condition is fulfilled, a rule will receive **4 points**.

## 02
### Range of values [2,3)

## Second Category

A rule will belong to this category if **all** its *LHS* items share at least one term (not necessarily the same) with at least one *RHS* item.

| A | B | | C | D |
|---|---|---|---|---|
| GO:01 | GO:04 | → | GO:04 | GO:09 |
| | GO:05 | | GO:01 | |

Each time this condition is fulfilled, a rule will receive **3 points**.

## 03
### Range of values [3,4)

## Third Category

A rule will belong to this category as soon as <u>one of its *LHS* items</u> share <u>one</u> term with one *RHS* item.

| A | B | | C | D |
|---|---|---|---|---|
| GO:01 | GO:02 | → | GO:02 | GO:09 |
| | GO:05 | | GO:03 | |

Each time this condition is fulfilled, a rule will receive 2 points.

## 04
### Range of values [4,5)

## Fourth Category

Here we can find all rules that, not fulfilling previous conditions, present at least one match between *LHS* items or between *RHS* items

| A | B | | C | D |
|---|---|---|---|---|
| GO:01 | GO:01 | → | GO:04 | GO:09 |
| | GO:05 | | | |

Each time this condition is fulfilled, a rule will receive 1 point.

## 05
### Range of values [5,6]

## Bottom Category

Rules that do not present any match will be allocated into this category.

**Fig 1. Assignable categories and ranking scores for a rule in the biological measures "BP", "MF", "CC" and "SP".** First, a rule will be assigned to a particular category from the bottom category 5 to the top category 1. This assignation will be conducted according to the type of matches encountered for a rule in its annotated terms as is described in the figure. Once a rule is designated to a particular category, a score will be computed for the rule taking into consideration all type of matches encountered for the rule. Each match is weighted with a number of points as illustrated in the figure. The final score for a rule is computed as detailed in the method section.

items, not always a gene in the LHS will be the cause of the change in the gene expression of a RHS gene (from a biological point of view). In many cases, rules could be just referring to a range of parallel phenomena that occur simultaneously at the gene level because of co-expression. Contrarily, in other cases, the method could be effectively representing true causal relationships between genes (e.g. rules 1 or 4). Precisely to discern these spurious associations from those true phenomena of interaction is why we propose the functional validation of the results and why some of the biological quality measures presented, such as the TF measure, become especially relevant.

Data visualization techniques have been widely employed in data science applications given their ability to transform model results into useful knowledge [43]. Beside their high ability to simplify big amounts of extracted information, graphs further allow information to be transferred in a very intuitive way to the user. In the context of high-dimensional data, such is the case of biological data, the problem of knowledge-extraction is much greater due to the large number of rules derived from the application of ML techniques [44]. Under these conditions, visualization techniques are presented as an attractive solution that serve as an interface between scientists and the extracted knowledge [45]. In this paper, we propose a new visualization tool that integrates output gene networks along with all accessed biological information. Based on hierarchical edge bundling methods [46], our tool generates circular plots illustrating the full picture of sequential rules discovered by our algorithm. In order to be extended to

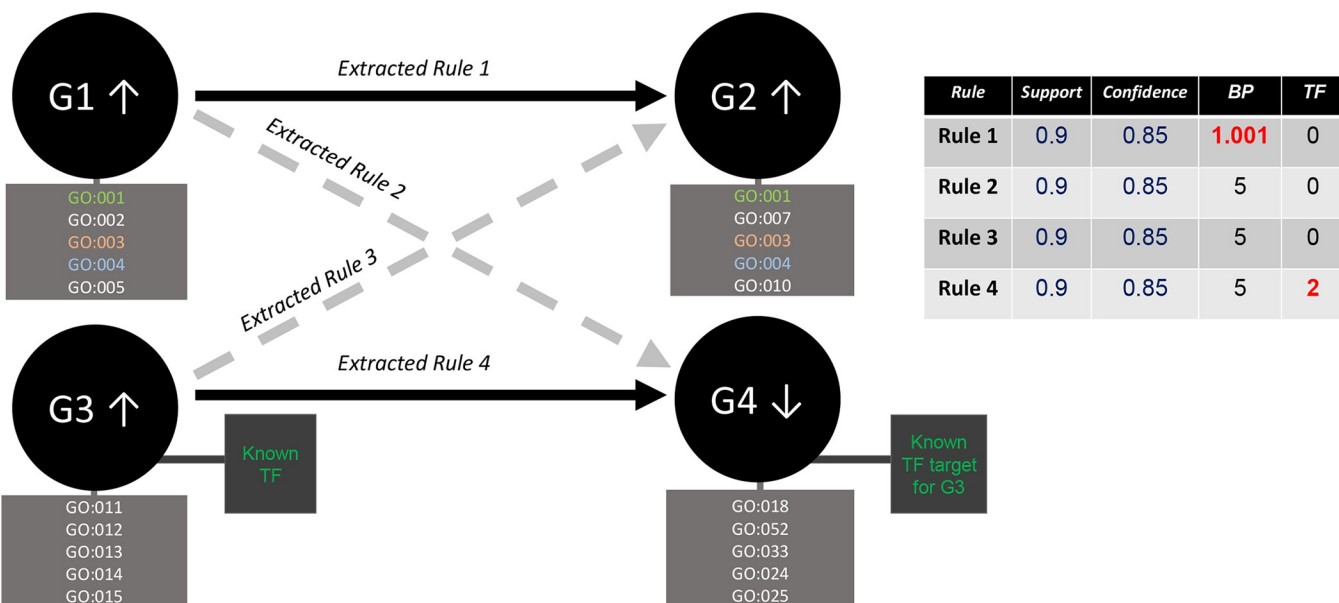

**Fig 2. Role of biological quality measurements for the functional assessment of each discovered gene-gene relationship.** While all extracted rules present acceptable and identical quality metrics (support = 90% and confidence = 85%), only the rule 1 presents a good BP measure value (remember that the range of values available for the BP measure was from 5 to 1, being the values near to 1 the ones corresponding to a higher number of GO matches between LHS and RHS genes). On the other hand, it is only the rule 4 the one presenting a good value for the TF measure (whose range of values was from 0 to 3, being 3 the maximal score for indicating a true TF-target gene relationship). The figure illustrate how the functional validation of results is critical to discern between spurious associations and true phenomena of interaction.

other temporal GED applications, this visualization method has been implemented as open-source software in R environment. By modifying circular diameters, edge width, edge type and the color intensity of each connection, the software generates plots comprising both the new biological quality measures and the classical SRM metrics computed by rule. Of note, our visualization tool is not restricted to the information presented in figures and can be easily adapted to each user demands. The greatest virtue of generated plots lays in their ability to concentrate different types of information in single shot, which further allows an easy identification of the top and more coherent rules from the both the technical and biological points of view. On summary, our visualization tool constitutes an additional value proposal in favor of model explicability and interpretability.

## Problem and datasets description: Long-term interventions in obesity

From the clinical or biomedical perspective, the real challenge issue when inferring gene networks is their reliability for avoiding false discovery as well as their reproducibility across different patient cohorts. For this reason, and given the lack of benchmark SRM methods, we decided to validate of our proposal in two alternative ways: 1) First, we applied our methodology to an example dataset (discovery sample) and give the derived results to a group of field-experts in order them to evaluate the usability of inferred networks for the generation of particular gene-gene interaction hypotheses; and 2) Second, we repeated the application of the full pipeline to three additional datasets, following the same experimental design than the discovery sample, and mined results looking for replication patterns across studies.

As an example of long-term human interventions, we chose a discovery dataset consisting of *in vivo* temporal GED derived from human adipose tissue (AT) samples collected in different time points during the course of a dietary intervention. With up to three time records available in the dataset, this study constitutes a perfect example of the *in vivo* temporal microarray experiments in which our method could extract biologically relevant gene-gene temporal relationships. Published by Vink et al. (2016) [47], the original clinical trial investigated the effects on weight loss (WL) of two different dietary interventions in 57 adults with obesity. Subjects were randomly assigned to each experimental group: a low-calorie diet (LCD; 1250 kcal/day) for 12 weeks (slow weight loss) or a very-low-calorie diet (VLCD; 500 kcal/day) for 5 weeks (rapid weight loss). In both experimental conditions, the WL period was followed by a 4-week additional phase of weight stabilization (WS). Abdominal subcutaneous AT biopsies were collected from each subject at each time point (baseline, after WL and after the WS period) and submitted to microarray analysis using the *Human Gene 1.1 ST* Affymetrix platform (one array per subject and time). A more detailed description of the study design can be found in the original publication of the dataset [48].

The full dataset was downloaded from the public repository GEO with identifier GSE77962. Fluorescence data were transformed into the form of an N x M matrix of gene expression values, where the N rows correspond to subjects under study and the M columns correspond to evaluated gene probes. The dataset presented valid fluorescence measures for 33.297 probes (M columns) mapping 19.654 unique genes across the genome. The number of individuals presenting valid gene expression data was 24 on the VLCD group and 22 on the LCD group. Since all available time records were merged into a single primary database, each individual presented three consecutive entries in the database (long format), corresponding to its gene expression profile at each temporal point (baseline, after WL and after the WS period). The final number of rows in the database was N = 138.

Data were normalized using RMA and submitted to feature-selection. The original number of probes was thus reduced to those DE genes by time interval and experimental condition. Time intervals corresponded to the WL period (comprising the end of WL vs baseline), the WS period (comprising the end of WS vs the end of WL) and the dietary intervention (DI) period (comprising the end of WS vs baseline). As a result, 431 probes matching 398 unique genes were selected for further analyses.

Remaining GED for the 431 probes were then submitted to data discretization. In each experimental group, a discretized matrix A of 431 probes was obtained, in which each probe by time interval may have one of three discrete states: 2, 1 and 0, meaning 'increase', 'decrease' and 'nochange' respectively. Once discretized, two sequence databases were constructed (one database per diet group) where each sequence corresponded to a subject and each event represented the change in the gene expression of a certain probe during a particular time interval (WL period or WS period). An example of the general structure of each constructed sequence database in the discovery case of study is presented in the S3 Fig. Details for each sequence database are presented in the caption of the figure.

Knowledge-extraction was conducted by CMRules in each experimental group separately and also by contrast (extracting only those association patterns exclusive for each experimental group). With the aim of avoiding losses of information, only results derived from the mining of each group separately were employed for functional validation. CMRules results in form of sequential rules were thus submitted to functional validation and the five biological quality measures were computed by rule and added to the already present five frequentist quality metrics. Finally, derived output were visually represented by means of our hierarchical edge bundling visualization tool.

Field-experts received sequential rules results in the form of both tables and figures with traditional quality metrics and biological quality measurements included. The evaluation and interpretation of sequential rules and graphs by field-experts was committed following a few foundations: 1) Rules were ordered according to SRM quality metrics and biological quality measures; 2) Rules with very low values for SRM quality metrics were removed according to the reference values presented in the first *method subsection*. For this prune, the concept of very strong association rules was taken into account; 3) Correlation analyses were conducted between quality metrics and biological quality measures for remaining rules in order to evaluate the ability of CMRules to extract biologically relevant patterns; 4) Identification, on the help of visual representations, of interesting sequential rules and generation of particular gene-gene interaction hypotheses; and 5) Exploration of most interesting hypotheses (either by accessing to the list of GO annotation terms matches or by performing intensive literature search). Field-experts were selected from the research group "CB12/03/30038", belonging to the Spanish research network CIBEROBN (Physiopathology of Obesity and Nutrition), Institute of Health Carlos III (ISCIII), Madrid, Spain.

In order to validate and contrast the insights derived from this discovery dataset, we further accessed temporal GED from WL interventions in three independent cohorts (GSE70529 [49], GSE35411 [50] and GSE103766 [51]). Dataset details and main characteristics of the technical validation process are presented in Table 1. Each validation dataset was processed and analyzed following exactly the same pipeline than the discovery population. Results and gene patterns discovered during the validation process are reported in a specific *result subsection*.Not restricted to the obesity field, our entire pipeline could be applied to any human long-term intervention with up to two experimental conditions (e.g. placebo and treatment).

**Table 1. Datasets details and problem description.**

| GEO Identifier | Design | Intervention Details | Time records available | Nº subjects (female/ male) | BMI at the beginning of the study | Age (years) | Sample tissue | Array Platform | Nº mined Strong ARs | Network representation |
|---|---|---|---|---|---|---|---|---|---|---|
| GSE77962 (LCD group)* [47,48] | Dietary Intervention | 1250 kcal/d during 12 weeks and a weight stable period of 4 weeks | 3 (baseline, after weight reduction and after weight maintenance) | 22 (12/ 10) | 28–35 kg/ $m^2$ | 51.8 ± 1.9 | Abdominal Subcutaneous Adipose Tissue | Affymetrix Human Gene 1.1 ST | 40 | Fig 3 |
| GSE77962 (VLCD group)* [47,48] | Dietary Intervention | 500 kcal/d during 5 weeks and a weight stable period of 4 weeks | 3 (baseline, after weight reduction and after weight maintenance) | 24 (13/ 11) | 28–35 kg/ $m^2$ | 50.7 ± 1.5 | Abdominal Subcutaneous Adipose Tissue | Affymetrix Human Gene 1.1 ST | 301 | Fig 4 |
| GSE70529 [49] | Dietary Intervention | Low-calorie diet of self-prepared foods for consecutive 5,10 and 15% weightloss | 4 (baseline, after 5, 10 and 15% weight reduction) | 9 (8/1) | 37.9 ± 4.3 kg/$m^2$ | 44 ± 12 | Abdominal Subcutaneous Adipose Tissue | Affymetrix Human Gene 1.0 ST | 551 | S5 Fig |
| GSE35411 [50] | Dietary Intervention | 1200 kcal/d during 3 months and a weight stable period of 4 weeks | 3 (baseline, after weight reduction and after weight maintenance) | 9 (6/3) | 42.7 ± 1.4 kg/$m^2$ | 40 ± 3.73 | Abdominal Subcutaneous Adipose Tissue | Affymetrix Human Genome U133 Plus 2.0 Array | 83 | S6 Fig |
| GSE103766 (WeightLosers group) [51] | Dietary Intervention + exercise counselling | 800–1000 kcal/d during 6 weeks and a less restrictive diet plan + exercise counselling for 12 months | 3 (baseline, after 5 months and after 12 months) | 6 (3/3) | 34.64 (0.7) kg/$m^2$ | 21–48 | Abdominal Subcutaneous Adipose Tissue | Affymetrix Human Genome U133 Plus 2.0 Array | 870 | S7 Fig |
| GSE103766 (WeightRegainers group) [51] | Dietary Intervention + exercise counselling | 800–1000 kcal/d during 6 weeks and a less restrictive diet plan + exercise counselling for 12 months | 3 (baseline, after 5 months and after 12 months) | 13 (9/4) | 34.65 (0.85) kg/$m^2$ | 20–45 | Abdominal Subcutaneous Adipose Tissue | Affymetrix Human Genome U133 Plus 2.0 Array | 70 | S8 Fig |

BMI and age data are presented as mean ± SEM, mean (SE) or rather as a range.

* Datasets employed as discovery population.

## Results

### Discovery approach in the case of study

As we previously explained, with the aim of illustrating the performance of our method on human long-term intervention data, we accessed and downloaded a discovery dataset composed of 57 subjects with obesity participating in a long-term dietary program [47,48]. The dataset consisted on temporal GED collected in three different time records during the course of two dietary interventions (VLCD and LCD). In this dataset, we sought to mine sequential rules with the form of [gene A↑, gene B↓] → (time delay) [gene C↑, gene D↑, gene E↑], that could illustrate the WL-induced gene regulatory responses of AT in obesity. Before the application of our SRM methodology to the dataset, all described pre-processing stages were

conducted. Once the data were properly formatted as described in *methods*, the data mining process was initiated. Specific minimum sequential support and sequential confidence thresholds were set by experimental condition during the knowledge-extraction stage (minSeqSup = 0.45 and minSeqConf = 0.4 for the VLCD group, and minSeqSup = 0.4 minSeqConf = 0.4 for the LCD group). Standard quality measures employed in SRM (lift, CF and conviction) were further computed in order to estimate the interestingness of each mined pattern. Beside conventional quality measures, the method also computed five new biological quality measures by rule based on external biological information (including functional and pathway annotations, and TF-target gene regulatory data). Through this strategy, interaction results were biologically pruned and placed within the context of already well-explored molecular systems [26,40,41]. Finally, the method applied a data visualization technique for the joint representation of output gene networks and all accessed biological information.

From the application of this pipeline to the discovery dataset, 50 output sequential rules were identified from the LCD group and 325 from the VLCD group (S1 and S2 Tables respectively). With up to seven times more number of rules mined from the VCLD group than from the LCD group, our results are in accordance with previous findings from Vink et al. (2016) [47], which showed a higher impact in the gene expression of AT elicited by a rapid and aggressive WL in comparison to the effects derived from a light but more prolonged WL. From all extracted rules, only very strong sequential rules were considered for further evaluation (SeqSup > minSeqSup, Not(SeqSup) > (1- minSeqSup) and CF > 0), which constitute a suitable framework to discard misleading rules. During the evaluation process, output sequential rules were biologically assessed by means of the five new biological quality measures and graphically represented in two circular plots (Figs 3 and 4). Main descriptive statistics for the extracted rules by experimental condition are presented in Table 2. In general, extracted rules presented robust values for all computed quality measures, which indicates a good performance of the algorithm during the gene association mining process. With robust quality metrics values, we refer to a minimum support higher than established threshold, a confidence higher to 0.8, a conviction value higher than 1, a lift higher than 1.1, and CF distinct of zero and as near as possible to 1 (see preliminary *method subsection*). According to mean values by group, slightly better metrics values were obtained for the VLCD than for the LCD, which probably was motivated on the higher impact elicited by this intervention in AT. In both groups, top rules (presenting higher values in the traditional quality measures) involved genes participating in molecular processes previously reported as part of the WL-induced AT response (e.g. mitochondrial function, angiogenesis, inflammation and lipid and glucose metabolism) [47,50] (S4 Fig). Of note, top sequential rules also presented good rates in the new proposed biological quality measures (Fig 5). Especially for the case of biological quality measures TF and BP, we showed significant correlations with the traditional quality metrics CF, conviction and confidence. This fact reflects a good performance of the knowledge-extraction process, where the best sequential patterns identified (from the ML perspective) are also the more biologically soundness. On the other hand, the fact of absence of correlation between some other biological quality measures and traditional metrics (Fig 5) reinforces the need for the functional validation of results. That is to say, although traditional metrics may indicate that some rules are good from the technical point of view, the biological information is not always what it could be expected.

In order to assess the biological utility of our gene networking strategy, obesity-field experts evaluated all extracted rules making use of the computed biological quality measures and the graphical representations as previously described (Figs 3 and 4). Since the most plausible mechanisms underlying gene regulation is the action of TFs on their target genes, the TF metric was the first measure employed by experts for filtering and evaluating output sequential

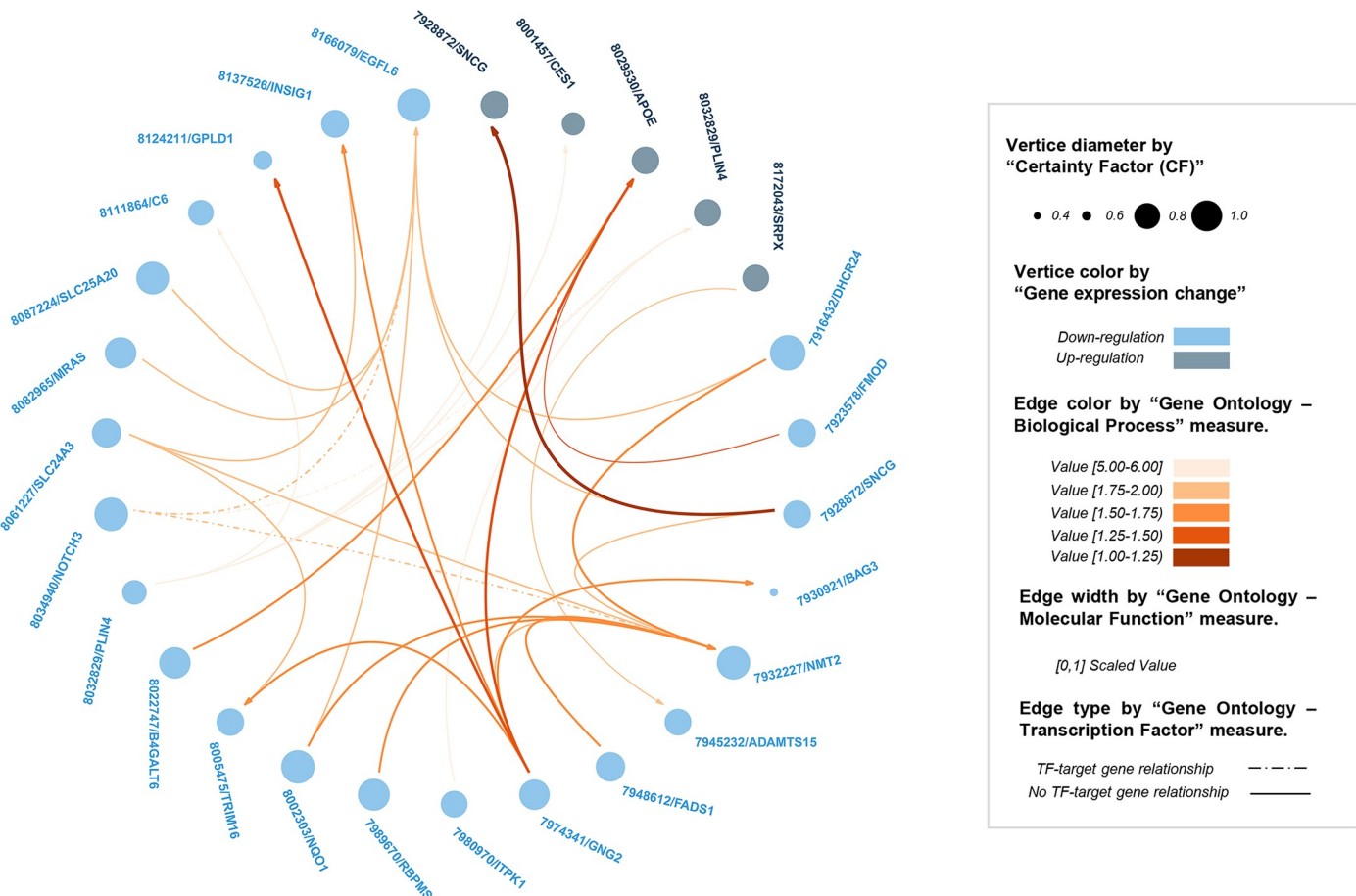

**Fig 3. Visual representation of the sequential rules discovered by our method in the GSE77962 dataset (LCD group).** Node names refer to (probe/gene).

rules. Through the application of a specific TF threshold ($> = 1$), four rules were selected from the LCD group and sixty-two from the VLCD group. Along with them, a subset of biologically meaningful rules are described in Table 3. From both intervention groups, several similar rules were identified sharing the same TF gene (*Notch3*) as an LHS item. In these rules, the gene *Notch3* emerged as a TF factor whose downregulation (provoked by the WL intervention) elicited later secondary changes in the expression of other genes during the WS period. Since each group was mined independently during the knowledge-extraction process, the fact of finding similar rules from each diet speaks well of the performance and the validation ability of our method. *Notch3* is mammalian transmembrane protein that bind membrane-bound ligands expressed by adjacent cells in human tissues. By triggering intracellular proteolytic cleavages and through the release of active intracellular domains of Notch (NICD), *Notch3* controls the expression of a wide range of target genes participating in different obesity-related processes such as differentiation, proliferation, angiogenesis and apoptosis. Interestingly, several *Notch3* target sequences have been identified within and near the genomic sequences of a few of its RHS genes (such is the case of *Nmt2* and *Clmn*) [52]. In these cases, sequential rules illustrate how a downregulation of the *Notch3* is followed by a downregulation and upregulation (respectively) of mentioned RHS genes. Despite these interesting results, it is important to clarify that the identification of sequential rules including a TF as LHS does not necessarily imply a causal relationship between the TF and its reported RHS gene. In these cases, functional *in*

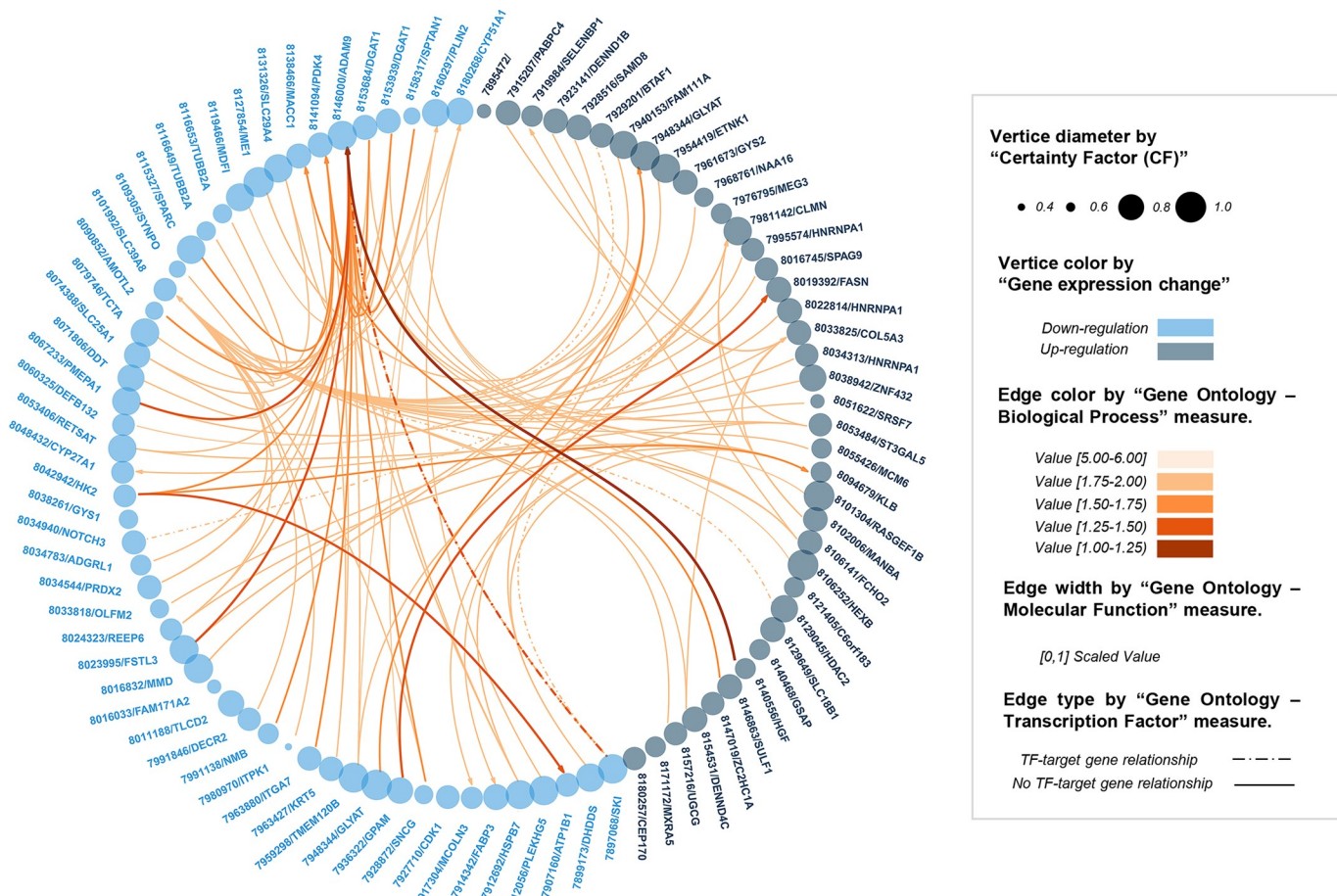

**Fig 4. Visual representation of the sequential rules discovered by our method in the GSE77962 dataset (VLCD group).** Node names refer to (probe/gene).

**Table 2. Descriptive statistics on quality metrics for strong association rules discovered in the whole GSE77962 dataset (LCD and VLCD groups).**

| | Support | Confidence | Lift | CF | VLCD Conviction | BP | CC | MF | SP | TF |
|---|---|---|---|---|---|---|---|---|---|---|
| *n* | 301 | 301 | 301 | 301 | 301 | 301 | 301 | 301 | 301 | 301 |
| *Minimum* | 11.00 | 0.71 | 1.13 | 0.22 | 1.27 | 1.001 | 1.001 | 1.001 | 1.2 | 0 |
| *Mean* | 11.08 | 0.88 | 1.55 | 0.71 | *Inf* | 2.19 | 1.94 | 2.17 | 3.8 | 0.21 |
| *Standard Dev.* | 0.27 | 0.09 | 0.23 | 0.22 | - | 1.1 | 0.48 | 1.06 | 2.4 | 0.41 |
| *Median* | 11.00 | 0.85 | 1.56 | 0.69 | 3.25 | 1.91 | 1.9 | 1.91 | 6 | 0 |
| *Maximum* | 12.00 | 1.00 | 2.00 | 1 | *Inf* | 6 | 6 | 6 | 6 | 1 |
| | | | | | LCD | | | | | |
| | [ort | Confidence | Lift | CF | Conviction | BP | CC | MF | SP | TF |
| *n* | 40 | 40 | 40 | 40 | 40 | 40 | 40 | 40 | 40 | 40 |
| *Minimum* | 9.00 | 0.69 | 1.08 | 0.27 | 1.36 | 1.002 | 1.002 | 1.002 | 1.2 | 0 |
| *Mean* | 9.72 | 0.81 | 1.38 | 0.54 | *Inf* | 2.39 | 1.77 | 2.71 | 4.56 | 0.1 |
| *Standard Dev.* | 1.26 | 0.09 | 0.17 | 0.18 | - | 1.55 | 0.16 | 1.8 | 2.23 | 0.3 |
| *Median* | 9.00 | 0.82 | 1.38 | 0.55 | 2.21 | 1.84 | 1.84 | 1.84 | 6 | 0 |
| *Maximum* | 15 | 1 | 1.69 | 1 | *Inf* | 6 | 1.96 | 6 | 6 | 1 |

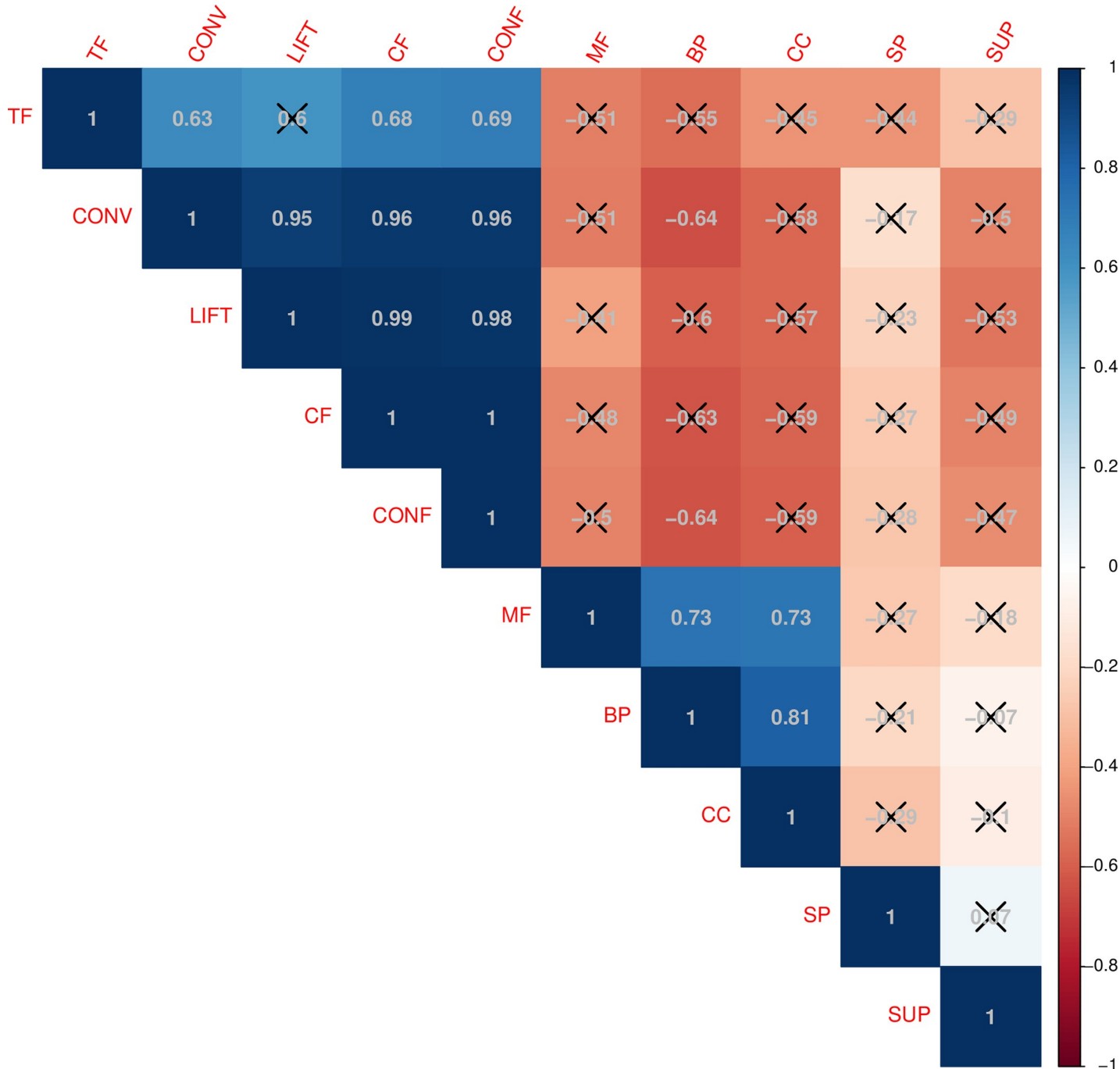

**Fig 5. Correlation between traditional quality metrics and biological quality measures by rule in the sequential rules discovered from the whole GSE77962 dataset (LCD and VLCD groups).** $R^2$ values quantify the level of correlation for each pair of measures while the level of statistical significance (adjusted by Bonferroni multiple test correction) is evidenced with an X for P-values > 0.05 and nothing for P-values < 0.05.

*vitro* studies should be performed for validating proposed interactions. From the LCD group it is also remarkable a rule with the *Notch3* as LHS and the *Egfl6* gene as RHS (Fig 3). Although no target sites for *Notch3* have been identified within the genetic sequence of *Egfl6*, a special functional connection has been evidenced between both genes in the context of obesity and

**Table 3. Subset of biologically meaningful extracted sequential rules in the whole GSE77962 dataset (LCD and VLCD groups).**

| Intervention Group | LHS | RHS | SUP | CONF | LIFT | CF | CONV | BP | MF | CC | SP | TF |
|---|---|---|---|---|---|---|---|---|---|---|---|---|
| VLCD | {8140556/HGF = 2} | {8146000/ADAM9 = 1} | 11.00 | 0.77 | 1.35 | 0.49 | 1.94 | 1.002 | 1.002 | 1.002 | 6.00 | 0.00 |
| VLCD | {7897068/SKI = 1} | {8146000/ADAM9 = 1} | 12.00 | 0.92 | 1.58 | 0.81 | 5.42 | 1.39 | 1.39 | 1.39 | 1.20 | 1.00 |
| VLCD | {8034940/NOTCH3 = 1} | {7981142/CLMN = 2} | 11.00 | 0.85 | 1.45 | 0.63 | 2.71 | 1.79 | 1.79 | 1.79 | 6.00 | 1.00 |
| LCD | {8034940/NOTCH3 = 1} | {8166079/EGFL6 = 1} | 12.00 | 0.86 | 1.11 | 0.37 | 1.59 | 1.79 | 1.79 | 1.79 | 6.00 | 1.00 |
| LCD | {7928872/SNCG = 1 & 8034940/NOTCH3 = 1} | {7932227/NMT2 = 1} | 9.00 | 0.90 | 1.52 | 0.76 | 4.09 | 1.83 | 1.83 | 1.83 | 6.00 | 1.00 |
| VLCD | {8131326/SLC29A4 = 1} | {8101992/SLC39A8 = 1} | 11.00 | 0.79 | 1.26 | 0.43 | 1.75 | 1.88 | 1.88 | 1.88 | 1.20 | 0.00 |
| VLCD | {8129045/HDAC2 = 2} | {8101992/SLC39A8 = 1} | 12.00 | 0.92 | 1.48 | 0.79 | 4.87 | 1.93 | 1.93 | 1.93 | 6.00 | 1.00 |
| VLCD | {7929201/BTAF1 = 2} | {8101992/SLC39A8 = 1} | 11.00 | 0.79 | 1.26 | 0.43 | 1.75 | 1.94 | 1.94 | 1.94 | 1.20 | 1.00 |
| VLCD | {7929201/BTAF1 = 2 & 7940153/FAM111A = 2} | {8101992/SLC39A8 = 1} | 11.00 | 0.92 | 1.47 | 0.78 | 4.50 | 1.94 | 1.94 | 1.94 | 1.20 | 1.00 |
| VLCD | {7929201/BTAF1 = 2 & 8106141/FCHO2 = 2} | {8101992/SLC39A8 = 1} | 11.00 | 0.85 | 1.35 | 0.59 | 2.44 | 1.95 | 1.95 | 1.95 | 1.20 | 1.00 |
| LCD | {8087224/SLC25A20 = 1& 8034940/NOTCH3 = 1} | {8166079/EGFL6 = 1} | 9.00 | 1.00 | 1.29 | 1.00 | Inf | 1.96 | 1.96 | 1.96 | 6.00 | 1.00 |
| LCD | {7980970/ITPK1 = 1 & 8034940/NOTCH3 = 1} | {8032829/PLIN4 = 2} | 9.00 | 0.75 | 1.50 | 0.50 | 2.00 | 6.00 | 6.00 | 1.90 | 6.00 | 1.00 |

angiogenesis [53,54]. In general, despite only a few previous evidences support an implication of the *Notch3* gene in obesity molecular pathways [55], the findings presented in this paper seem to point this TF as an important element for the proper regulation of AT cellular responses to WL.

For investigating the sequential rules discovered in the VLCD group (Fig 4), the use of other biological quality measures instead of TF (such as BP and MF) allowed experts to identify several interesting patterns. On the one hand, it highlights a sequential rule involving the loci *Fasn* (Fatty Acid Synthase) and the *Gpam* (Glycerol-3-Phosphate Acyltransferase 1, Mitochondrial); both of them genes coding for enzymes with a central role in the process of lipogenesis. The sequential rule between these genes was easily identified from its color intensity in the circular plot and may suggest a special relevance of the lipogenesis process as part of the responses of obese AT after a strong caloric restriction. Another interesting insight extracted from the graph is the fact that most of the gene expression changes elicited by the dietary intervention in the VLCD group ended in a later and secondary downregulation of the gene expression levels of *Adam9* during the WS period. Among all sequential rules illustrating this behavior, there are a few ones with special biological relevance (Fig 4); one highlighted by the BP metric and involving the gene *Hgf*, and another one including a TF-target gene regulatory relationship with the protooncogene protein *Ski*. *Adam9* is a cell-surface metalloprotease present in almost all cells and tissues of the body that participates in key processes such as cell migration, proliferation and cell-cell interactions. Mostly expressed by white cells, *Adam9* has been reported to get upregulated during many pathological processes including cancers. Regarding obesity, previous transcriptomics analyses have demonstrated how *Adam9* is significantly up-regulated in obese AT and how it plays a major mediating role in a chain of interactions that connect local inflammatory phenomena to the alteration of AT metabolic functions [56,57]. On this sense, the downregulation of Adam9 evidenced in our study might constitute a biologically meaningful finding with relevance for the understanding of the AT metabolic

health amelioration achieved with dietary intervention in this case of study. All quality metrics for the sequential patterns highlighted in this section have been resumed in Table 3, while the full list of sequential rules identified can be explored in S1 and S2 Tables. Taking all these into consideration, the model and its output results were considered by field-experts as an easily interpretable approach that could be successfully extended to other human long-term intervention datasets for the identification of biologically relevant molecular signatures.

## Validation approach in independent cohorts

In order to validate and contrast the insights derived from the discovery dataset, we accessed additional temporal GED from three WL interventions performed in independent cohorts (GSE70529 [49], GSE35411 [50] and GSE103766 [51]). Dataset details and main characteristics of each population are presented in Table 1. Although there were slight differences in the study design of each cohort, all studies constituted dietary interventions (caloric restriction programs) performed during a long-term intervention period in adult subjects with obesity. Each experimental group (in the case of datasets presenting more than one study condition) was again considered as an individual dataset. During the knowledge-extraction stage, minSeqSup = 0.5 and minSeqConf = 0.6 thresholds were set and only "very strong rules" were selected for subsequent evaluation. In Table 1, we report the number of very strong association rules mined from each dataset. Visual representations of output gene patterns by dataset are presented in S5, S6, S7 and S8 Figs. Graphs illustrated again coherent gene-gene interactions within the context of obesity research (e.g. the gene association patterns governed by the locus *Abca1* reported in the dataset GSE70529 (S5 Fig)) [58]. These figures display information following the same code of colors and format than previously presented Figs 3 and 4.

Very strong rules extracted from all datasets were pulled together for the identification of replicated patterns. During the process of contrasting rules between datasets, probe information was removed from each rule and only the locus tag of each item was considered. That is to say, we considered two sequential rules as replicates when they contain the same genes within LHS and RHS (but not necessarily the same probes). As a result, we found gene expression changes in 11 loci acting as trigger mechanisms (LHS items) concurrently in sequential rules extracted from different datasets (these were *C6* = Up-regulation, *Hnrnpa1* = Up-regulation, *Srsf7* = Up-regulation, *Gsap* = Up-regulation, *Sncg* = Downregulation, *Notch3* = Downregulation, *Srpx* = Up-regulation, *Itpka1* = Downregulation, *slc-transporters* = Downregulation, *Tmem-proteins* = Downregulation and *Znf-proteins* = Up-regulation). Interestingly, these validated trigger loci included TFs, splicing factors, mRNA processing molecules and cell surface transporters with a great implication in the control of the global gene expression cell profiles [59–62]. In the same manner, we found the gene expression change of 1 loci represented as a consequence (RHS item) in several ARs extracted from different datasets. This gene expression change corresponded to a downregulation of the locus *C6*, which encodes a component of the complement cascade with implication in the innate immune system and inflammation pathways. Among all extracted rules, those containing at least one of the described common LHS and RHS loci were selected for further evaluation (S3 Table). The graphical representation of all these rules allowed the identification of very interesting gene patterns and replicated interactions which have been shown in S9 Fig. On the one hand, we found rules from different datasets illustrating a sequential association between the downregulation of *Slc-transporter* genes and a subsequent downregulation of proteins from the *Adam*-family (S9A Fig). In the same manner, we replicated a sequential relationship between the gene expression change of *Tmem* genes and the later modification in the expression of loci from the *Srsf*-family (S9A Fig). Particularly, while a relationship of the type (down-regulation -> down-regulation) was

found between these genes in the weight losers of the dataset GSE35411, a relationship of the type (up-regulation–> up-regulation) was found between these genes in the GSE103766 dataset (a cohort composed of weightregainers) (S9A Fig). The target loci of these rules corresponded to serine/arginine-rich (SR) proteins, a conserved RNA-binding protein family, which consists of 12 members, serine/arginine-rich splicing factor (SRSF)1-12 in humans [60]. SR proteins have demonstrated multiple key roles in the control of gene expression, including constitutive and alternative pre-mRNA splicing, transcription, mRNA transport, mRNA stability and translation [60]. Therefore, these genes could perfectly be key regulatory points through which the WL intervention elicit long-term changes in adipocytes. Beside these replicated patterns, during the investigation of the set of rules containing common LHS or RHS (S3 Table), we also noticed a rebound effect in the gene expression of certain loci during the dietary intervention program (S9B Fig). Particularly, we observed how although certain genes experimented a downregulation of their gene expression in response to WL, these genes returned to their original gene expression status as soon as a normal-calorie diet was restored (exhibiting some kind of negative and positive feedback loop regulations of their own expression, which could be the explanation of fast transient dynamic changes or the maintaining in time of their expression levels). Altogether, these validated patterns might represent the sequence of genetic changes that occur in AT during a long-term weight loss intervention. Indeed, some of the identified loci have already been drawn as key genes or targets for the management of many complex diseases [62].

## Discussion

Temporal gene networking has emerged as an effective approach for filling the missing heritability gap of complex human traits. Until date, several ML approaches have been proposed for the dynamic modelling of time course omics data, highlighting co-expression clustering methods [1]. Although they have yielded impressive results in terms of model accuracy and predictive ability, most of these applications are based on "Black-box" algorithms and more interpretable models have been claimed by the research community [10]. Especially during the reconstruction of gene networks, one of the main concerns of biologists has been how to translate inferred networks into particular hypotheses that can be tested with real-life experiments. Fortunately, the recent XAI revolution offers a solution for this issue [63–65], were rule-based approaches are highly suitable for explanatory purposes [16,17]. Within this context, SRM approaches have emerged as an interesting XAI method for the modelling of temporal gene-gene interactions *in vitro* [15]. Some of the best characteristics of SRM methods for this task include the existence of statistical quality measures by interaction, the possibility of biological validation by relationship, the inclusion of time (causality) order information in networks or their ability to discover complex regulatory phenomena. Taking all these into account, and given the fact that temporal co-expression clustering methods present some drawbacks as described earlier, we propose that SRM could serve as an alternative of great interest and interpretability for mining particular temporal relations between genes in humans. The further integration of the data mining process along with functional annotation and pathway resources is an additional way towards more explanatory and biologically soundness models [4].

In the present study, we propose a full pipeline for extracting sequential rules from temporal GED through the application of SRM in longitudinal microarray human studies. As far as we concern, this is the first application of a naturally interpretable method for the modelling of temporal gene-gene relationships in humans. The whole pipeline of our method is illustrated in the S1 Fig. Gathered under open-source software, our proposal could be extended to any

temporal GED human study, with special applicability in long-term interventions or clinical trials. The presented pipeline is organized on three main blocks:

1. Data Pre-processing stage, involving feature-selection and data discretization.

2. Knowledge-extraction stage, consisting of the adaptation of the algorithm CMRules to the problem of temporal GED.

3. Functional validation of results, in which we propose five new biological quality measures by rule and a tool for visualizing the results.

The two strategies adopted during the data pre-processing stage were intended to deal with some of the well-known human omics data complexities. As evidenced in the case of study, both strategies resulted useful for increasing model interpretability and for reducing the search space into a high quality data subset. During the second phase of the approach (knowledge-extraction), an SRM algorithm was adapted to the temporal GED problem given the previously proven ability of rule mining methods for extracting biologically meaningful gene association patterns both in static [14] and dynamics datasets [15]. Particularly, a method known as CMRules was chosen as a good technique for this task. CMRules implementation was accomplished following published recommendations in gene association analysis [14] and the biological knowledge played an important role during the mining process.

With this work, we have tried to move away from the "black-box" concept that is adopted in many of the current AI omics highthroutput applications, in which complex genetic networks are extracted from datasets without obtaining useful knowledge for the experts. An example of this kind of "poorly explanatory" models is the work recently published by Tareen et al. (2018) [66], with one of the datasets employed here. Although some interesting gene networks are reported in the work, the output format of co-expression networks and their visual representation are poorly explainable by itself, especially for the generation of particular hypotheses of gene-gene interactions. Moreover, the approach lacks of a method for the functional validation of established gene-gene relationships, thus hindering the biological interpretation of results.

In contrast, our approach presents a high *eXplainability*, which is mainly achieved by two consecutive ways: 1) given the type of employed knowledge-extraction algorithm, and 2) thanks to the third proposal of the pipeline, which includes the functional validation and visual representation of results.

Regarding the knowledge-extraction algorithm, the chosen SRM method CMRules constitutes a methodological advance in comparison to previous SPM approaches. It greatest virtue emanates from the format in which its results are presented. This is the form of rules: X -> (time delay) Y, where each interaction between two or more genes could be suggesting a causal time-lagged relationship between them. For example, sometimes, these interactions could be indicating how the increase in the amount of a TF causes a subsequent increase or decrease in the expression of a target gene, while other times they could suggest how two distinct genes (participating in a same metabolic route) increase their expression consecutively after an intervention. In the latter case, for example, the interaction would be illustrating how the activation of certain biological pathways is maintained over time in response to an intervention, after a first trigger event. Additionally, in other cases, when it is the same gene the one that occupies both LHS and RHS positions, rules could be suggesting negative or positive feedback phenomena (which could serve as explanation for fast transient dynamic changes or the maintaining in time of expression levels of certain genes in long-term interventions). Interestingly, all introduced types of relationships have been reported in our tested datasets (see results section) and assert with the two core ideas of XAI:

- Explainable models, while maintaining a high level of learning performance (e.g. support, confidence, conviction, CF, lift).

- Enabling human users to understand, appropriately trust, and effectively manage the emerging generation of artificially intelligent methods.

On the other hand, *eXplainability* is also achieved in our pipeline with the creation of five new biological quality measures by rule and by the visual representation of results. Although previous association rule mining studies have already employed the GO and KEGG resources for the functional validation of extracted rules [14], this is the first time specific biological quality measures by rule are computed taking into consideration external molecular knowledge. The creation of biological quality measures by rule constitute an ideal initiative to sort and explore identified gene patterns according to any desired biological criterion. Moreover, it is also the first time that TF-target gene regulatory information has been taken into account for the functional interpretation of gene rules. Since TF gene regulation is the most plausible molecular mechanism underlying gene-gene and gene-environment interactions, this initiative could be extended to any other gene association analysis. Finally, although the results obtained directly from the CMRules algorithm in the form of sequential rules are perfectly interpretable themselves, their exploration from a global perspective may be somewhat complicated. For this reason, we though that the evaluation of these rules including their biological quality information would greatly benefit from a visual group approach, where all the rules could be studied together. Thus, we decided to incorporate our visualization tool as a stage for the functional validation of the results. We chose a type of plots whose greatest virtue lays in their ability to concentrate different types of information in single shot, which would allow practitioners an easy identification of the top and more coherent rules from the both the technical and biological points of view. On this matter, our visualization tool it is not intended as a network analysis tool (for generating networks from raw data) but only an additional value proposal in favor of model explicability and interpretability.

From the analyzed datasets, computed biological quality measures and the visualization tool demonstrated utility for the biological interpretation of results and the transference of large gene patterns to the expert eye. Thanks to them, field-experts were able to identify several rules corresponding to known biological relationships among genes. Moreover, although our CMRules algorithm does not strictly output a network-format like result such as co-expression approaches do, when one visualizes all sequential rules at a single shot, It is evidenced how SRM interestingly keep the scale-free network topology for inferred interactions. This network topology, which is also evidenced by co-expression approaches results, it is in tune with the concept of "good enough solutions" that seems to rule most biological systems and it consists on the existence of a few nodes with many connections ("hubs") and many nodes with few connections [18]. This, again, demonstrates the suitability of our SRM approach alongside the visualization tool for the modeling genetic interactions in humans.

Given the absence of a gold-standard which to compare with our approach, and taking into account that the important thing when inferring genetic networks in the biological field is to validate results in independent cohorts, in this work we decided to validate our proposal through its direct application to different cases of study. Within the context of the chosen research problem ("WL interventions in obesity"), this is the first study implementing a XAI analytic approach in temporal gene networking. Moreover, by incorporating data from up to 4 independent cohorts and 6 experimental groups (N = 83 subjects), this analysis also constituted one of the biggest omics applications in the field of omics interventions. After applying our whole pipeline on these datasets, we not only identified interesting gene networks within each of the mined datasets but also validated some of the patterns primarily extracted from the

discovery sample. Altogether, these results further reinforce the goodness of our strategy for the mining of biologically relevant gene-gene temporal relations under different conditions and clinical designs. An exhaustive study of all the results from the case of study is needed otherwise to understand the concrete molecular patterns underlying WL-induced responses in obesity. In order to make this pipeline analysis extensible to any other temporal GED, we have implemented all described methods in open-source scripts and the codes have been shared online. Our method is not necessarily restricted to microarray data and could also be extended to RNAseq and other NGS technologies. For example, for applying our methodology to RNA-seq datasets, it would be enough by simply counting on an *N X M* gene expression matrix of normalized reads. In future works, it would be of great interest testing our approach with RNAseq expression datasets given the stronger reliability of these data in term of the technical robustness of sequencing platforms. Approaches like this would greatly expand our knowledge for complex biological processes, with a special interest for long-term intervention experiments (such as clinical trials), in which gene regulatory mechanisms could reveal new drug targets.

The high dimensionality of microarray data is a permanent problem for this kind of approaches. For future analyses, it would also be advisable to test the effect of employing different "feature selection" and "discretization" strategies on the performance of the algorithm. In addition, it would be convenient that the biological quality measures could be computed at the same time that the rule extraction process, in such a way that they can guide the method within the search space. As a result, methods will be able to find fewer rules but with higher biological quality, which may otherwise remain hidden.

Finally, future works could also be focused on improving the computing of biological quality measurements based on GO ontology terms. For that purpose, we will combine our heuristic approach alongside the available tools that have been developed to evaluate the biological similarity of two genes based not only on the identical GO terms that they share, but also on the rest of GO terms that are annotated (not identical) [42]. In the future, a combined approach like this could be of great interest to improve the functional validation of our method and will be taken into consideration for the continuation of the work. Besides this modification, other future approaches could also consist of performing the visual representation of the rules with ontology terms representing nodes instead of genes. This would allow us to visualize networks in terms of functionality and to understand how cellular functions follow each other in human tissues after long-term interventions. In this case, the difficulty would be to identify which GO terms are the most characteristic for each gene in order to represent them within the network. Once achieved, the way in which the nodes of the network are connected could be different to our current representations and thereby reveal novel information extracted by the method that is not observed with our current approach.

## Code availability

All data manipulation and processing steps as well as all secondary statistical GED analyses were conducted in R environment using the next list of libraries (*"Matrix"*, *"lattice"*, *"fdrtool"*, *"rpart"*, *"affy"*, *"oligo"*, *"affydata"*, *"ArrayExpress"*, *"limma"*, *"Biobase"*, *"Biostrings"*, *"genefilter"*, *"affyQCReport"*, *"affyPLM"*, *"simpleaffy"*, *"ggplot2"*, *"dplyr"*, *"pd.hugene.1.1.st.v1"*, *"FGNet"*, *"RGtk2"*, *"RDAVIDWebService"*, *"topGO"*, *"KEGGprofile"*, *"GO.db"*, *"KEGG.db"*, *"reactome. db"*, *"org.Hs.eg.db"*, *"arules"*, *"arulesViz"*). All employed codes have been gathered under a unique pre-processing R script, which is available online. The implementation of CMRules was carried out in Java using the open-source data mining library "SPMF" (http://www. philippe-fournier-viger.com/spmf/). The computing process for the five new biological quality

measures was implemented in Python version 3.7 (http://www.python.org). The data visualization process instead was implemented in R environment. The codes for running all described processes (pre-processing, CMRules mining, computing of biological quality measures and the data visualization tool) have been shared online and can be easily extended to any other application. This software is distributed as open source software under the terms of the GNU Public License GPLv3 and it is hosted in the public hosting GitHub (https://github.com/AugustoAnguita/GeneSeqRules).

## Supporting information

**S1 Fig. Description of the proposed pipeline for the inference of sequential gene-gene interactions (pre-processing, knowledge-extraction and functional validation stages).** The values (0,1 and 2) from the discretization step correspond to ('No change in gene expression','-Downregulation' and 'Upregulation). In order to simplify the figure, the data from the example only represent gene expression values from one individual in the population. Thus, and since our discretization approach uses the mean SLR (from all individuals in a dataset) for computing discrete states, they should not be intended as an example of a real discretization process in our approach.
(TIF)

**S2 Fig. Example of the execution of the CMRules algorithm.** This figure is a modification of the Fig 3 available in the original publication of CMRules [25].
(TIF)

**S3 Fig. Structure of the input sequence databases required by CMRules.** Each database must contain discrete temporal information (events) for changes in gene expression. Events were represented using a 4-digit code (where the first digit represents the change in gene expression (1 = Downregulation, 2 = Upregulation, NA = no change) and the next three digits represent the identifier for the *i* probe under study). Section A refers to the sequence database constructed in the GSE77962 dataset (VLCD group) and B to the sequence database constructed in the GSE77962 dataset (LCD group).
(TIF)

**S4 Fig. Functional enrichment analysis of the genes included in sequential rules discovered from the whole GSE77962 dataset (LCD and VLCD groups).**
(TIF)

**S5 Fig. Visual representation of the sequential rules discovered by our method in the GSE70529 dataset.**
(TIF)

**S6 Fig. Visual representation of the sequential rules discovered by our method in the GSE35411 dataset.**
(TIF)

**S7 Fig. Visual representation of the sequential rules discovered by our method in the GSE103766 dataset (WeightLosers group).**
(TIF)

**S8 Fig. Visual representation of the sequential rules discovered by our method in the GSE103766 dataset (WeightRegainers group).**
(TIF)

**S9 Fig. Highlights among the sequential rules with common LHS and RHS loci discovered in the validation datasets.** The complete set of sequential rules that contain at least one of the identified key LHS and RHS loci (from the validation study) is available in the S3 Table. A refers to validated patterns between datasets while B illustrates rebound effects in the gene expression of certain loci during the dietary intervention program evidenced in different datasets.
(TIF)

**S1 Table. 50 output sequential rules identified from the discovery GSE77962 dataset in the LCD group.**
(PDF)

**S2 Table. 350 output sequential rules identified from the discovery GSE77962 dataset (VLCD group).**
(PDF)

**S3 Table. The complete set of sequential rules that contain at least one of the identified key LHS and RHS loci (from the validation populations).**
(PDF)

## Acknowledgments

The authors would like to thank the owners of datasets (GSE77962, GSE70529, GSE3541 and GSE103766) and the GEO repository for making science accessible for everyone and for encouraging the open data culture. The authors also acknowledge the University of Granada "Plan Propio de Investigacion 2016-Excellence actions: Unit of Excellence on Exercise and Health (UCEES)". This paper will be part of Augusto Anguita-Ruiz's doctorate, which is being completed at the University of Granada, Spain.

## Author Contributions

**Conceptualization:** Augusto Anguita-Ruiz, Rafael Alcalá, Jesús Alcalá-Fdez.

**Data curation:** Augusto Anguita-Ruiz.

**Formal analysis:** Augusto Anguita-Ruiz.

**Funding acquisition:** Augusto Anguita-Ruiz, Rafael Alcalá, Concepción M. Aguilera, Jesús Alcalá-Fdez.

**Investigation:** Augusto Anguita-Ruiz.

**Methodology:** Augusto Anguita-Ruiz, Rafael Alcalá, Jesús Alcalá-Fdez.

**Project administration:** Jesús Alcalá-Fdez.

**Software:** Augusto Anguita-Ruiz, Alberto Segura-Delgado.

**Supervision:** Rafael Alcalá, Concepción M. Aguilera, Jesús Alcalá-Fdez.

**Validation:** Augusto Anguita-Ruiz.

**Visualization:** Augusto Anguita-Ruiz.

**Writing – original draft:** Augusto Anguita-Ruiz.

**Writing – review & editing:** Augusto Anguita-Ruiz, Rafael Alcalá, Concepción M. Aguilera, Jesús Alcalá-Fdez.

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
