## [Decision Letter · Decision Letter 0]

17 Oct 2019

Dear Dr Anguita-Ruiz,

Thank you very much for submitting your manuscript 'eXplainable Artificial Intelligence (XAI) for the identification of biologically relevant gene expression networks in longitudinal human studies.' for review by PLOS Computational Biology. Your manuscript has been fully evaluated by the PLOS Computational Biology editorial team and in this case also by independent peer reviewers. The reviewers appreciated the attention to an important problem, but raised some substantial concerns about the manuscript as it currently stands. While your manuscript cannot be accepted in its present form, we are willing to consider a revised version in which the issues raised by the reviewers have been adequately addressed. We cannot, of course, promise publication at that time.

Sincerely,

Benjamin Althouse

Associate Editor

PLOS Computational Biology

Alice McHardy

Deputy Editor

PLOS Computational Biology

[LINK]

Reviewer's Responses to Questions

**Comments to the Authors:**

Reviewer #1: review uploaded as attachment

Reviewer #2: This paper tackles the problem of explainable artificial intelligence for omics data. As such, the topic is currently very interesting as most existing techniques are black-boxes, hence, they are not interpretable for clinicians, biologist, etc.

The authors use a novel rule-based strategy that includes preprocessing, knowledge extraction and biological validation to tackle the problem. They have run experiment on in-vivo GED in 57 subjects with obesity.

Overall, the paper is very well-written, it is easy to follow and the contribution is clearly relevant for a great audience. In addition to that, I am really happy to see that the authors are sharing all the source code generated on Github.

I only have some minor suggestions:

- As the case study is very much focused on obesity, perhaps the title should also reflect that..as this pipeline might not always work in GED.

- Data preprocessing seems to be focsed only on discretisation, which is definitely key. Just wondering if the authors could discuss other forms of data preprocessing. For example, feature reduction (selection) may be very interesting with this kind of data.

- It is clear that the results are interpretable, and the figures and table reported are proof of that. However, I would like to see a bit of additional discussion about this; in particular, would this approach always be interpretable? what are the caveats to use this approach? Maybe good to discuss potential future work from here.

Reviewer #3: - Naming the step where rules are overlaid in biological databases “Biological validation” may be misleading. What the authors refer in this step will be more in the framework of Functional validation. Please rephrase accordingly in the main text, supplementary material and code.

- Rules for scoring:

Please explain how the authors decided on the scoring system for each category. Are they purely heuristic? Have the author evaluated the effects of varying the assigned values?

For matching terms of LHS and RHS, are the inherent term ontologies in the databases considered? E.g. would a child-ancestor pair be considered a hit? Please explain why yes/no.

- Discretization is a key aspect in the presented methodology and merits deeper explanations. In the main text is stated:

“- For probes showing a positive signal log ratio:

If log2(FoldChange)ikj > log2(FoldChange)ikJ Then assign the label ‘Upregulation’.

Otherwise type ‘No change’

- For probes showing negative signal log ratio:

If log2(FoldChange)ikj < log2(FoldChange)ikJ Then assign the label ‘Downregulation’.

Otherwise type ‘No change’

Where the term log2(FoldChange)ikj refers to the change in the gene expression for the i probe, in the k time interval and the j subject, and the term log2(FoldChange)ikJ otherwise refers to the mean signal log ratio for that particular ik probe in all subjects from the group under study (our designated data scope).”

a) For log2(FoldChange)ikJ I believe the authors are referring to mean signal log ratio for a particular probe i at a given time point k in all subjects from the group under study J. Please clarify.

b) Probes need to be defined in the sentence “probes showing a positive/negative signal log ratio”. I believe they refer to a probe ikj (probe, timepoint, subject)

c) If I’m correct in b) I can think of several cases where down/up regulation events are missed out. For example for a given sign for the log ratio where the group has inverted signed. Also when the log ratio is positive but significantly lower than the group (downregulated). The opposite case, when the log ratio is negative but significantly higher than the group (upregulated). Please clarify.

- Please explain in Supp Fig 1. what 1,2, or 0 means in the Discretization step.

- As NGS techniques such RNA-seq are now standard with a variety of applications, the paper will benefit by mentioning if or how the proposed methodology can be applied.

I particularly enjoy that the source code is made available and shared. I think that some more work is needed in the usability of the methodology in order to allow the scientific community to use this novel approach:

- As much as possible avoid hard-coded filenames in all steps. One can use Rscript to run R with command line arguments.

- As much as possible provide a simplified execution: Pipeline steps which are feeding directly from the output of the previous step or can run sequentially. For example step 1, 2, and 3.

- For step 3 avoid asking the user to manually prepare the format. The instructions can be rather easily done programatically.

- Avoid as much as possible asking for having superuser rights (step 4). This is very limiting if working in shared computing facilities. If it's a must, then a virtual environment may help.

- Minor: In the source code "required _codes": rename to src or just place them in the root folder.

**Have all data underlying the figures and results presented in the manuscript been provided?**

Reviewer #1: Yes

Reviewer #2: Yes

Reviewer #3: Yes

PLOS authors have the option to publish the peer review history of their article (what does this mean?). If published, this will include your full peer review and any attached files.

Reviewer #1: No

Reviewer #2: No

Reviewer #3: No

---

## [Decision Letter · Decision Letter 1]

17 Mar 2020

Dear Mr Anguita-Ruiz,

We are pleased to inform you that your manuscript 'eXplainable Artificial Intelligence (XAI) for the identification of biologically relevant gene expression patterns in longitudinal human studies, Insights from obesity research.' has been provisionally accepted for publication in PLOS Computational Biology.

Best regards,

Benjamin Althouse

Associate Editor

PLOS Computational Biology

Alice McHardy

Deputy Editor

PLOS Computational Biology

Reviewer's Responses to Questions

**Comments to the Authors:**

Reviewer #2: The authors have properly addressed my previous comments. In my opinion this paper can be accepted in its current form.

Reviewer #3: The authors have addressed all my comments satisfactory.

**Have all data underlying the figures and results presented in the manuscript been provided?**

Reviewer #2: Yes

Reviewer #3: Yes

PLOS authors have the option to publish the peer review history of their article (what does this mean?). If published, this will include your full peer review and any attached files.

Reviewer #2: No

Reviewer #3: No

---

## [Editor Report · Acceptance letter]

2 Apr 2020

PCOMPBIOL-D-19-01444R1 

eXplainable Artificial Intelligence (XAI) for the identification of biologically relevant gene expression patterns in longitudinal human studies, Insights from obesity research

Dear Dr Anguita-Ruiz,

I am pleased to inform you that your manuscript has been formally accepted for publication in PLOS Computational Biology. Your manuscript is now with our production department and you will be notified of the publication date in due course.

With kind regards,

Matt Lyles
